# Spiking attractor model of motor cortex explains modulation of neural and behavioral variability by prior target information

Vahid Rostami [1,4], Thomas Rost[1,4], Felix Johannes Schmitt [1], Sacha Jennifer van Albada [1,2], Alexa Riehle [2,3] & Martin Paul Nawrot [1] ✉

When preparing a movement, we often rely on partial or incomplete information, which can decrement task performance. In behaving monkeys we show that the degree of cued target information is reflected in both, neural variability in motor cortex and behavioral reaction times. We study the underlying mechanisms in a spiking motor-cortical attractor model. By introducing a biologically realistic network topology where excitatory neuron clusters are locally balanced with inhibitory neuron clusters we robustly achieve metastable network activity across a wide range of network parameters. In application to the monkey task, the model performs target-specific action selection and accurately reproduces the task-epoch dependent reduction of trial-to-trial variability in vivo where the degree of reduction directly reflects the amount of processed target information, while spiking irregularity remained constant throughout the task. In the context of incomplete cue information, the increased target selection time of the model can explain increased behavioral reaction times. We conclude that context-dependent neural and behavioral variability is a signum of attractor computation in the motor cortex.

Despite correct motor performance in well-trained animals, behavioral reaction times and neural spiking activity in the motor cortex appear highly variable across repetitions of an identical task[1]. Hence, any successful biologically realistic model of cortical motor control must provide a mechanistic explanation for both types of variability.

Over the past decades, attractor dynamics has been established as the most viable mathematical concept and computational model to support working memory and decision-making in sensory-motor tasks, and there is considerable experimental evidence to support it[2–10]. Yet, despite the fact that the basic theory of attractor dynamics is well understood[11–13], its robust implementation and functional application in biologically realistic spiking neural network models remains a challenge.

The classical cortical model is the balanced network of excitatory and inhibitory neurons with random connectivity[14,15]. This model successfully captures certain aspects of cortical spiking dynamics, including low spontaneous firing rates and irregular spiking statistics. However, it fails to explain the observed cortical trial-to-trial variability and cannot accommodate metastable attractors. Metastable attractors (or simply metastability) refers to the phenomenon where the neural dynamics exhibit transient states of coordinated activity across ensembles of neurons, known as metastable states. These states

[1]Institute of Zoology, University of Cologne, Cologne, Germany. [2]Institute for Advanced Simulation (IAS-6), Jülich Research Center, Jülich, Germany. [3]UMR7289 Institut de Neurosciences de la Timone (INT), Centre National de la Recherche Scientifique (CNRS)—Aix-Marseille Université (AMU), Marseille, France. [4]These authors contributed equally: Vahid Rostami, Thomas Rost. ✉e-mail: martin.nawrot@uni-koeln.de

persist for durations ranging from tens to hundreds of milliseconds before transitioning to the next state. The presence of such metastable sequences is widespread across various cortical areas and behavioral contexts, prompting investigations into their functional role in sensory and cognitive processing[9,16–20]. Each metastable state is characterized by a specific pattern of ensemble activity, involving the firing rates of a collection of neurons. The transitions between these states are marked by abrupt, jump-like modulations, presenting neural activity as a sequence of discrete and quasi-stationary states. This phenomenon is observed in response to external stimuli as well as during spontaneous activity.

Over recent years, the balanced random network has been extended to accommodate strongly interconnected neuron clusters. With carefully tuned network parameters, this architecture introduces metastable attractor dynamics and can capture aspects of cortical trial-to-trial variability dynamics[16,17,21–23]. These studies used purely excitatory neuron clustering but neglected any structure in the topology of local inhibitory networks.

Despite the vital role of inhibitory neurons in cortical dynamics, the circuit connectivity of inhibitory neurons has remained poorly understood until recent years. Several influential studies provided evidence that inhibitory interneurons connect non-specifically to surrounding excitatory pyramidal cells[24,25], which inspired the term "blanket of inhibition" and supported the rationale behind a purely excitatory cluster topology[21,22,26]. However, additional studies now provide a more complete picture, suggesting a high degree of specificity and possible clustering of inhibitory neurons[27–34]. In particular, it has been argued, based on anatomical and physiological evidence, that inhibitory networks can be strongly interconnected locally. Moreover, neurons that receive strong excitatory input typically also receive strong inhibitory input, supporting local balancing at the level of single cell input[27,35,36]. In addition, recent theoretical studies have corroborated the importance of inhibition in attractor type dynamics and found, based on the analytical treatment of binary and firing rate models, that inhibitory clustering strongly improves the robustness of metastable attractor dynamics[33,37].

In the present study, we propose a network architecture for spiking cortical attractor networks using combined excitatory and inhibitory clustering. We show that inhibitory clustering maintains the local balance of excitation and inhibition and yields the desired metastability robustly over a wide range of network sizes and parameters. We utilize our model to mechanistically explain task-related dynamics of multiple single-unit activity recorded from the motor cortices in two monkeys (M1 and M2) during a delayed reaching task. Here we consider the decision-making process and do not attempt to generate musclebah-like signals to generate movement trajectories. We find that our model qualitatively and quantitatively captures in vivo firing rates, task epoch related dynamics of trial-to-trial variability and spiking irregularity, and behavioral reaction times. Variation of the behavioral task involved different levels of target uncertainty during the delay period and resulted in corresponding levels of neural trial-to-trial variability and the systematic variation in behavioral reaction times, both in the monkey data and in our model simulations.

## Results

### Spiking networks with local excitatory-inhibitory clusters can explain cortical variability dynamics in vivo

We start out with analyzing the temporal dynamics of spike train variability in single-unit recordings from the monkey motor cortex during a delayed center-out reach task[38,39] (see Methods). The monkeys were instructed to reach for one of six target buttons at the end of an instructed delay period during which a varying degree of target certainty was cued by either one, two or three adjacent targets. We first consider the simplest task condition in which complete target information was provided to the monkey with the onset of the preparatory period (preparatory signal, PS, indicated by the green circle in Fig. 1a). After a fixed delay of 1 s the monkey was prompted to execute the movement by the response signal (RS, red circle). Correct execution of a trial was rewarded with fruit juice.

Common statistical measures for quantifying the variability dynamics of spiking neurons are the Fano Factor (FF) and the local coefficient of variation (CV$_2$) of inter-spike intervals[40–42]. The FF determines the neural response variability across the repetition of trials while the CV$_2$ measures the variability of the inter-spike intervals and thus quantifies the irregularity of spike occurrences within each trial. On theoretical ground, the expectation values for both measures are interdependent (see Methods) and their empirically estimated absolute values and relation can be interpreted with respect to the nature of the underlying generative neuronal and network processes (see Discussion). We apply FF and CV$_2$ in a time-resolved fashion using a sliding observation window[41] (see Methods). Fig. 1a shows that during spontaneous activity and before cue onset (PS) trial-to-trial variability is high (FF ≈ 1.8). After PS, the FF decreases significantly (red curve in Fig. 1a) before reaching a constant plateau. This task-related reduction of the FF has been demonstrated previously in motor cortices and for different behavioral tasks[39,43–45] as well as in other cortical areas[44,46]. The irregularity of inter-spike intervals, CV$_2$, on the other hand, remains constant over time (black curve in Fig. 1a) and does not show any dependency on the experimental epochs. For the Poisson process the expectation is FF = CV$_2$ = 1 (dashed gray line in Fig. 1a).

Next, we study variability dynamics in a biologically plausible spiking network model of the cortex with excitatory cluster topology. Deviating from the random balanced network, the excitatory neuron population is divided into subpopulations with strong internal connectivity while excitatory connections between clusters are comparatively weak[16,21–23]. The model is composed of 4000 excitatory (E) and 1000 inhibitory (I) exponential integrate-and-fire neurons. The E neurons are organized into $Q = 50$ recurrent clusters (Fig. 1b top panel). Synaptic connections within each excitatory cluster are potentiated by a factor $J_{E+}$, and, to maintain overall balance, connections between E neurons belonging to different clusters are depressed by a factor $J_{E-}$ (see Methods). Inhibitory neurons are non-specifically connected to themselves and to E neurons (see Methods). A consistent reduction in FF upon stimulation can only be achieved with a strong stimulus (dark red curve) whereas we observe an inconsistent behavior for intermediate stimuli or even an increase in FF for stimuli of low strength (light red curve), conflicting with our experimental observation. As for the dynamics of irregularity, we observe a reduction in CV$_2$ during stimulation that tends to be stronger for a stronger stimulation. This stimulus-induced increase in regularity is again clearly inconsistent with our experimental observation (Fig. 1a) and, to our knowledge, has not been reported in any other experimental study. In summary, the E-clustered network is inconsistent with the experimental observations in two ways: (1) a reduction in count variability (FF) is achieved only with a strong stimulus, while a weak stimulus can lead to an increase in FF; (2) during network stimulation the irregular spiking is disrupted, and the CV$_2$ assumes unrealistically low values.

To match the experimentally observed stimulus-induced variability dynamics, we here suggest the following type of network connectivity. Recent anatomical and physiological studies point to a high local connectivity and a possible clustering of inhibitory neurons[27–29,31–33]. We therefore combine excitatory and inhibitory clustering in our spiking network model following our previous proposal for binary networks[37] (Fig. 1c top panel). We simulate this E/I-clustered network model using the same parameters as for the E-clustered network (see Methods). Figure 1c shows that the trial-to-trial variability (FF) decreases robustly even for a weak network stimulation while the CV$_2$ does not show any stimulus-induced changes, matching our experimental observations (Fig. 1a). We will next

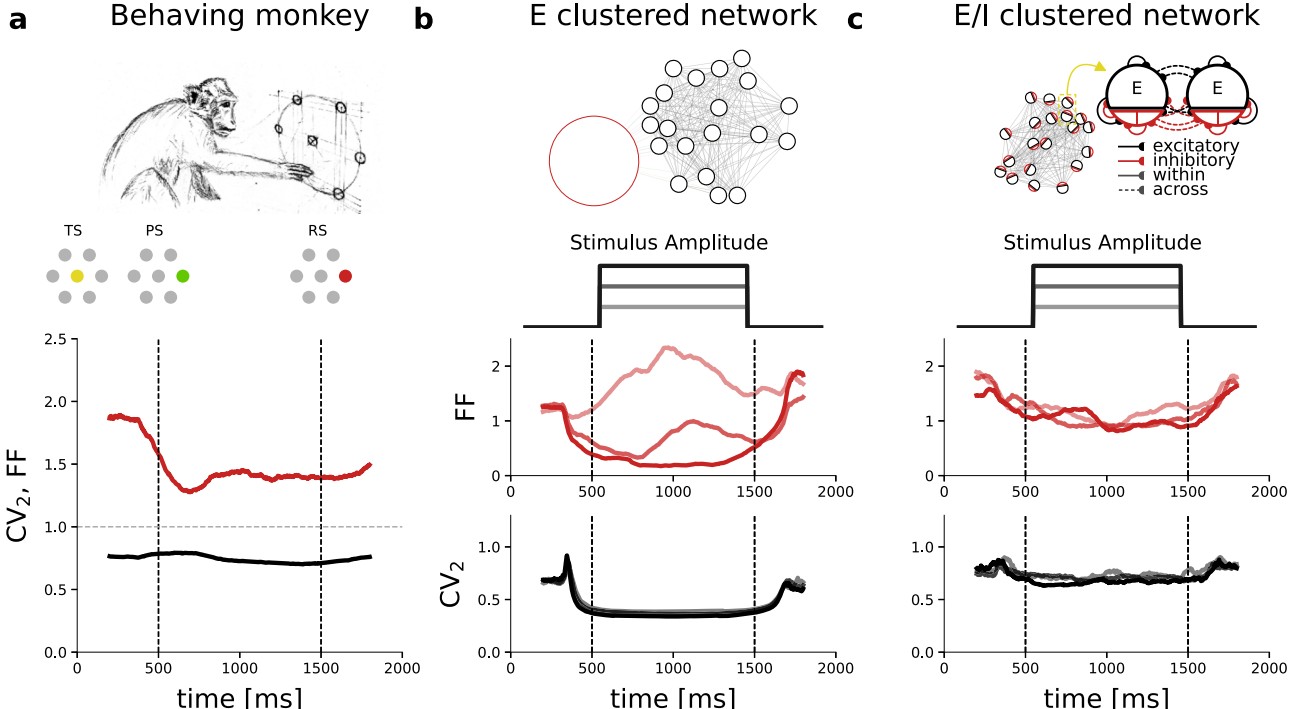

**Fig. 1 | Variability dynamics observed experimentally and in clustered spiking network models. a** Experimental data recorded from the motor cortex of M1 during a delayed center-out reach task[39]. After the monkey places its hand on the central touch-sensitive LED the trial start (TS) is indicated by lighting up the LED (yellow). At $t = 500$ ms the preparatory stimulus (PS) indicates the movement target (target LED is lit in green). After a fixed preparatory delay period of 1000 ms, during which the monkey is not yet allowed to move his hand, the response stimulus (RS) is indicated by changing the color of the target LED to red, and the monkey is allowed to move. The time-resolved FF and $CV_2$ averaged across neurons are shown in red and black, respectively. Gray horizontal dashed line shows FF and $CV_2$ for the Poisson process. **b** Network model with purely excitatory clusters and global inhibition (E-clustered network). Clusters in the network receive step-wise external input with three different stimulus amplitudes (0.2 pA, 0.25 pA, 0.3 pA) as indicated in shades of gray. The corresponding time-resolved FF is shown in shades of red and the time-resolved $CV_2$ in shades of gray. **c** Same analyses as in (**b**) but for the proposed network model with excitatory and inhibitory clusters (E/I-clustered network). FF and $CV_2$ were computed in a 400 ms sliding window; Note that we use a centered window and therefore the curve can start decreasing before the indicated trigger (max 200 ms); simulated data comprised 50 trials.

investigate the network mechanisms that support realistic variability dynamics in E/I-clustered but not in E-clustered networks.

## Metastability emerges robustly in the E/I-clustered spiking network

We first ask which network parameters determine the emergence of metastability and winnerless competition among embedded clusters. To answer this question we examined the effect of two important clustering parameters, the clustering strength $J_{E+}$ and the number of clusters $Q$ for a fixed network size. Metastability is expressed in the successive activation and inactivation of individual neuron clusters where individual neurons switch between lower and higher firing rate states. Across repeated observations of individual neurons we therefore expect a high variation of the spike count. Thus, the FF provides a proxy measure for metastability.

We again consider a network of 4000 excitatory and 1000 inhibitory neurons (Table 1). We first kept the number of clusters $Q = 50$ fixed and varied the cluster strength between $J_{E+} = 1$, which coincides with the classical random balanced network without clustering, to $J_{E+} = Q = 50$ with zero excitatory coupling between different clusters (see Methods). The E-clustered network (Fig. 2a) can show metastability as indicated by the alternating firing rate states of individual cluster populations (insets in Fig. 2a) only in a very narrow range of excitatory cluster strengths around $J_{E+} \approx 3.5$, which is accompanied by correspondingly large Fano Factors (FF > 1). When increasing $J_{E+}$, the cluster dynamics rapidly breaks down and the FF falls below its initial value associated with the random balanced network ($J_{E+} = 1$) and eventually the network gets stuck in a single state with a few clusters

becoming permanently active with high firing rates and regular spiking patterns.

A different picture emerges in the E/I-clustered network (Fig. 2c). Metastability is achieved over a wide range of $J_{E+}$. With increasing

**Table 1 | Standard network parameters for the simulations of the E/I-clustered spiking neural network simulations**

| Parameter | Unit | Value |
|---|---|---|
| $N$ | - | 4000($E$); 1000($I$) |
| $E_L$ | mV | 0 |
| $V_{th}$ | mV | 15 |
| $V_R$ | mV | 0 |
| $C_m$ | pF | 1 |
| $\tau_m$ | ms | 20($E$); 10($I$) |
| $\tau_{syn}$ | ms | 3($E$); 2($I$) |
| $\tau_r$ | ms | 5 |
| $p_{EE}$ | - | 0.2 |
| $p_{EI}, p_{IE}, p_{II}$ | - | 0.5 |
| $g$ | - | 1.2 |
| $J_{EE}$ | pA | 0.25 |
| $J_{EI}$ | pA | −0.66 |
| $J_{IE}$ | pA | 0.19 |
| $J_{II}$ | pA | −1.00 |
| $I_x$ | pA | $2.13I_{th}(E)$; $1.24I_{th}(I)$ |

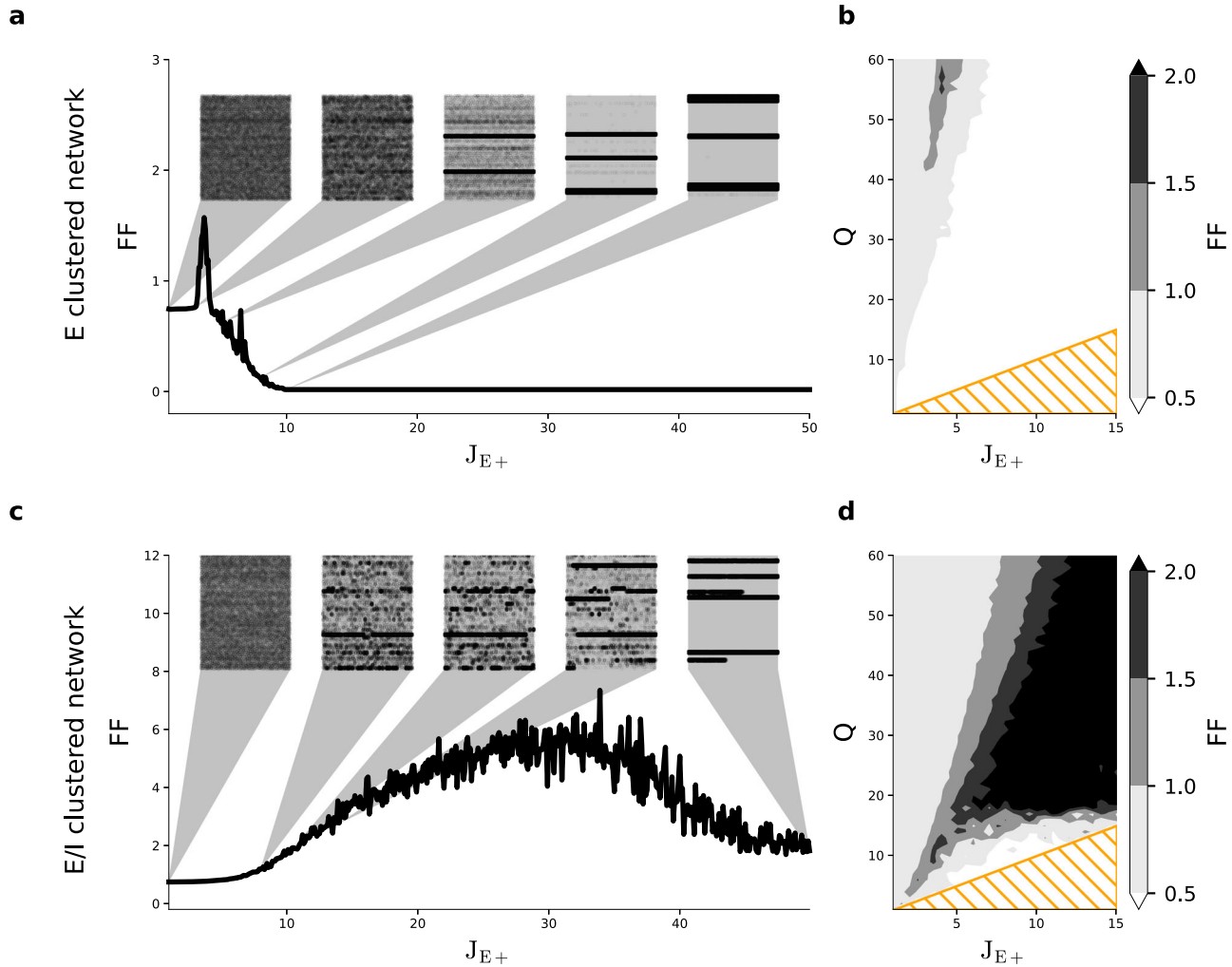

**Fig. 2 | Excitatory-inhibitory clustering facilitates winnerless competition across a wide range of network parameters. a**, **c** FF versus $J_{E+}$ for networks with $Q = 50$ clusters computed from 20 trials of 400 ms duration and averaged over 50 network realizations. Insets show 2 s of single trial network activity for all excitatory neurons grouped by their cluster identity. **b**, **d** Effect of cluster strength $J_{E+}$

and number of clusters $Q$ on metastability and FF. Shaded orange triangles indicate the zone below $J_{E+} = Q$, where the clusters are completely decoupled. **a**, **b** Show results for the E-clustered network (see also Supplementary Fig. 1); (**c**, **d**) for the E/I-clustered network. Parameters apart from $J_{E+}$ and $Q$ as in Table 1. Insets in (**a**) for $J_{E+} = (1, 3, 3.7, 8, 10)$, in (**c**) for $J_{E+} = (1, 8, 10.5, 14, 50)$.

cluster strength, the duration of individual cluster activations becomes longer. For large values $J_{E+} \gtrsim 15$ the variance over repeated simulations is high, as some cases exhibit extensive cycling between clusters while in other cases one or a few clusters become dominantly active and suppress the winnerless competition dynamics. At $J_{E+} = Q = 50$ coupling between the populations exists only through strong inhibitory connections and most populations are silenced by a few winning clusters. The desired attractor dynamics thus takes place at lower values of $J_{E+}$, which yield realistic average trial-to-trial variability in the approximate range $1 < FF < 3$.

In a next step we in addition varied the number of clusters $Q$ while keeping the total number of neurons $N$ fixed. This changes the size of each cluster, i.e., larger $Q$ means a smaller number of neurons per cluster. For the purely excitatory cluster topology (Fig. 2b) we find that there is only a very small parameter region that shows the desired high spike count variability across repeated observations (FF > 1). Thus, the E-clustered network is able to facilitate winnerless competition, but stable switching dynamics can be achieved only by extensive parameter tuning. Litwin-Kumar and colleagues[21] suggested a variation of their E-clustered network architecture where each excitatory cluster preferentially excites one disjunct pool of neurons within the total inhibitory population. Repeating our calibration for this alternative

architecture, we again obtained the same result (Supplementary Fig. 1) as for the E-clustered network (Fig. 2b). In contrast, in the E/I-clustered network attractor dynamics as indicated by a high FF > 1 emerges robustly over a large range of cluster numbers $Q$ and excitatory cluster strengths $J_{E+}$ as shown in Fig. 2d. This means that metastability in the E/I-clustered network is not sensitive to variations or perturbations in network parameters. Importantly, the E/I-clustered topology supports metastability even for large clusters in contrast to the E-clustered network where attractor dynamics breaks down for larger cluster sizes (lower values of Q).

## Local balance of excitation and inhibition facilitates attractor dynamics and maintains spiking irregularity

Synaptic excitation and inhibition are opposing effects that together determine the activity and maintain the excitability of cortical neurons and networks[35,47,48]. A number of physiological and model studies have shown that excitatory and inhibitory synaptic inputs retain a fixed and roughly equal proportion during both spontaneous and evoked states[36,49–53]. This mechanism known as balance of excitation and inhibition is crucial for maintaining irregular spiking and stable network activity, and disruption of this balance has been linked to pathological activity states such as epileptic seizures[52].

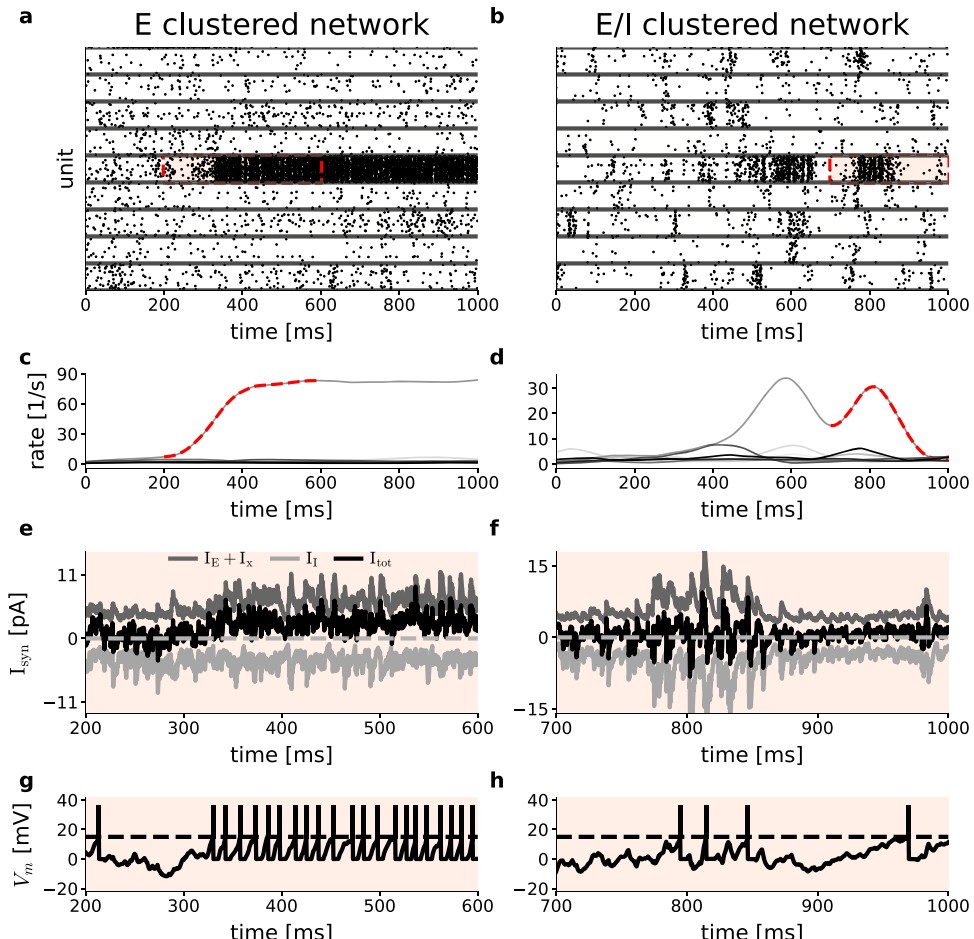

**Fig. 3 | E/I-clustering maintains local balance of excitation and inhibition and irregular spiking.** Comparison of E-clustered (left) and E/I-clustered (right) networks. **a**, **b** Spike raster display of nine excitatory clusters during 1000 ms of spontaneous activity. Horizontal lines separate different clusters. Red shaded area indicates the epoch of interest where a switch into an active state occurred in one cluster. **c**, **d** Average firing rate of each of the nine clusters shown above as estimated with a 50 ms triangular kernel[104]. Dashed lines correspond to the red shaded region of interest in the upper panels. **e**, **f** Synaptic currents in a randomly chosen unit around the switch to the active state within the cluster and epoch of interest as indicated above. **g**, **h** Membrane potential for the unit shown in (**e**) in the same time interval.

In this section, we study the balance of excitation and inhibition in E- and E/I-clustered networks. Using again the parameters in Table 1 we simulate 1000 ms of spontaneous activity of E- and E/I-clustered networks with $Q = 50$ clusters. Figure 3 illustrates the difference between E-clustered (left panels) and E/I-clustered (right panels) networks on the level of individual neurons. The raster plot in Fig. 3a shows the activity of nine sample excitatory clusters in the E-clustered network. Neurons across all nine clusters exhibit low firing rates until at about $t = 350$ ms when one cluster switches into an activated state. The corresponding instantaneous firing rate averaged across all neurons in the activated cluster increases strongly to almost 90 spikes/s, whereas the firing rates of the other clusters remain low. Figure 3e shows the synaptic input currents to one single neuron in the activated cluster around the switching time (red shaded interval in Fig. 3a). While the excitatory input current increases ($I_E$, upper trace) due to the strong mutual excitation within this cluster, the inhibitory input ($I_I$, lower trace) remains constant. As a result, the net input current ($I_{tot}$, middle trace) increases to positive values and hence the neuron operates in the mean-driven rather than in the fluctuation-driven regime. As a result, a large portion of the neurons in the active cluster fire with high rates and high regularity, as can be seen in the membrane potential of the example neuron depicted in Fig. 3g. This provides the mechanistic explanation for our results of a strongly decreased $CV_2$ in Fig. 1b.

The right-hand side of Fig. 3 analyzes the equivalent scenario for an E/I-clustered network. The raster plot in Fig. 3b indicates that in this spontaneous state individual clusters can switch from an inactive state with low ongoing firing rates to activated states of variable duration. This attractor dynamics involves moderate firing rates during activated states (Fig. 3d), which are considerably lower than the high firing rates observed in the E-clustered network Fig. 3d. The synaptic input currents in a sample neuron are shown in Fig. 3f, again at the transition from the inactivated to the activated state of its cluster. In effect, inhibitory clustering ensures that the inhibitory synaptic currents increase in parallel with the excitatory synaptic currents during cluster activation, thereby increasing the variance but not the mean of the net synaptic input, which is kept balanced throughout. Hence, all neurons remain in the fluctuation-driven regime and retain irregular spiking as illustrated in Fig. 3h and shown for the network population in Fig. 1c. In conclusion, maintaining balance of excitatory and inhibitory inputs to lower firing rates in activated clusters supports switching between inactive and active states by decreasing the rate differences of both, facilitating the desired metastable dynamics. In purely excitatory clusters the unbalanced input together with the strong connectivity among neurons within a cluster results in high firing rates of the excitatory neurons that operate in a mean-driven input regime. The increased self-excitation within an excitatory cluster and the lack of balancing inhibition leads to longer duration of an activated cluster

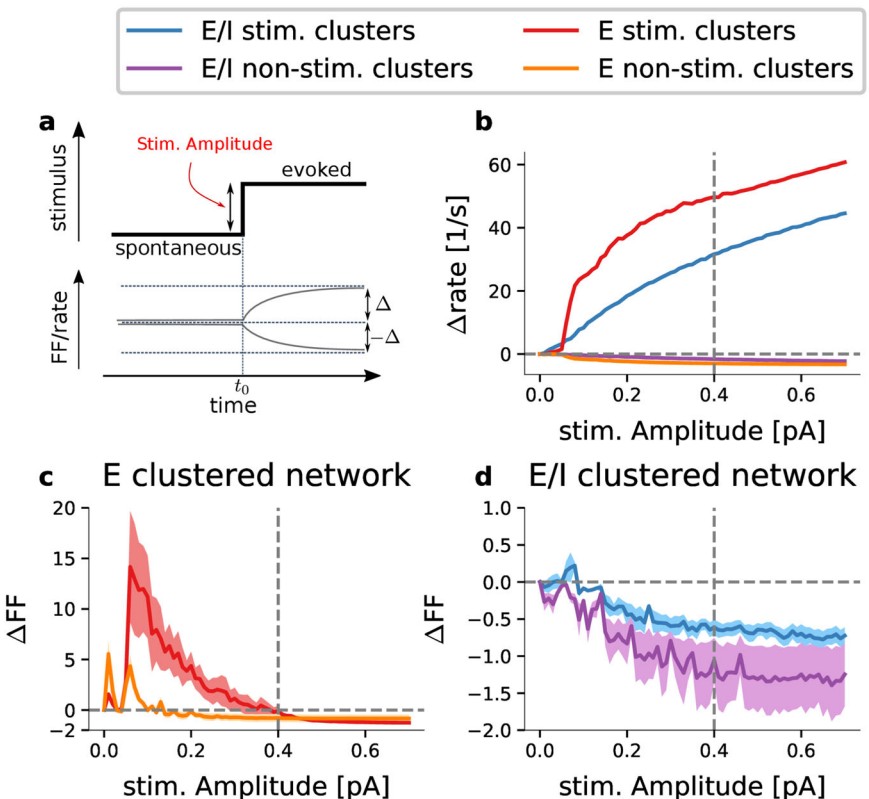

**Fig. 4 | Stimulus-evoked changes in firing rate and trial-to-trial variability.**
**a** Cartoon depicting the stimulus-induced transition from the spontaneous to the evoked cluster state. Upper panel shows the step change in the external input current at time $t_0$. Lower panel depicts the stimulus-induced changes in the average firing rate or FF relative to the spontaneous level. Stimulation is applied to 5 out of the total of $Q = 50$ clusters, leaving 45 non-stimulated clusters. **b** Average stimulus-response function Δrate versus the amplitude of the injected step current. Only periods of time within the average cluster rate that exceed the median rate of the non-stimulated clusters are included. Different colors indicate the behaviors for stimulated and non-stimulated clusters in the E and E/I-clustered networks as indicated in the legend. **c**, **d** ΔFF versus stimulus amplitude for the E-clustered network (**c**) and the E/I-clustered network (**d**) estimated across 50 trials. Both measures are calculated for 1000 ms of spontaneous and evoked activity, respectively. The difference is computed for each neuron separately before averaging across neurons. Firing rate differences of single units were averaged across the 50 trial repetitions. Shaded areas show the standard error of the mean calculated across the stimulated and non-stimulated clusters, respectively. Network parameters as in Table 1.

state and can result in permanent winner-takes-all scenario where a single cluster remains activated for a very long duration (Fig. 3a).

## E/I-clustering increases the dynamic range of the stimulus-response function

We have seen that the E-clustered network fails to capture the reduction in trial-to-trial variability during stimulation with a weak stimulus (Fig. 1). Here we ask what exactly constitutes a weak stimulus by analyzing in detail how the cluster response rate and the change in trial-to-trial variability depend on the stimulus amplitude in networks with and without inhibitory clustering.

To this end we stimulated five out of 50 clusters by means of a constant input current injected into all neurons belonging to these clusters (Fig. 4a). In our analysis we compare network activity in the spontaneous state (before stimulation) with the network activity during the evoked state (during stimulation) and calculate the changes in firing rate (Δrate) and in trial-to-trial variability (ΔFF) as a function of stimulus strength. In Fig. 4b we show the relation of stimulus amplitude and average response of the activated clusters separately for the neurons in the stimulated and in the non-stimulated clusters. In the stimulated clusters of the E/I network we observe a smooth relation between input current and firing rate increase across the tested stimulus range. In contrast, for the case of a purely excitatory cluster topology the firing rate in stimulated clusters increases steeply within a narrow range of stimulus current amplitude (≈0.05–0.08 pA) and continues to increase rapidly with increasing input current. The non-stimulated clusters exhibit a small decrease in their average firing rate during stimulation.

The dynamics of trial-to-trial variability again shows a clear difference in the two different network topologies. In the E-clustered network, there is a stark increase in FF for stimulus amplitudes in the lower half of the stimulus range of up to ~0.4 pA (Fig. 4c). The desired effect of a reduced FF is achieved only for increasingly large input currents that correspond with high cluster rate responses of Δrate > 40 spikes per second (cf. vertical dashed line in Fig. 4b, c). This means that the E-clustered network fails to reproduce the experimentally observed reduction in trial-to-trial variability for weak stimuli and moderate average neuronal response rates. The reason for the increase in variability for weak to moderate stimulus amplitudes is that stimulated clusters switch into an activated state in some trials but fail to do so in others, i.e., cluster activation is unreliable. This coincides with our theoretical prediction based on the analytic treatment and numerical simulation of binary cluster networks where stable fixed points require considerable excitatory input stimulation and can assume only high cluster rates[37].

In contrast, the E/I topology shows a reduction in trial-to-trial variability of single-neuron spiking in stimulated and non-stimulated clusters even for weak stimuli and correspondingly low average cluster response rates (Fig. 4b). This reflects that cluster stimulation reliably initiates switching into an excited state in each single trial. The range of ΔFF in the models closely matches the average ΔFF observed in the experimental data (cf. Fig. 1a). We will show that, for matching the

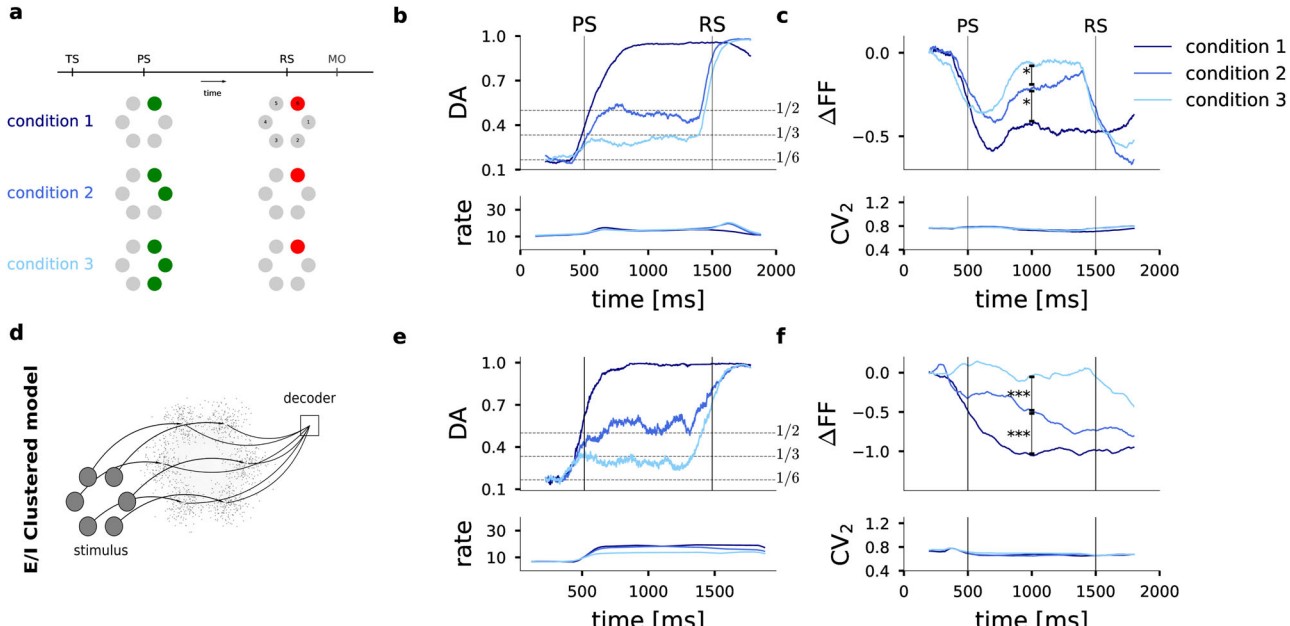

**Fig. 5 | Functional E/I-clustered network model captures context dependency of information encoding, firing rates and variability dynamics. a** Full experimental protocol of the delayed center-out reach task with three experimental conditions. The preparatory signal (PS) indicated either one, two or three adjacent targets in green, corresponding to Condition 1 (full target information), and Conditions 2 and 3 (incomplete target information). After a fixed delay period of 1000 ms the reaction signal (RS) resolved target ambiguity by indicating the final movement target in red, prompting the monkey to execute the movement. The movement onset time (MO) and movement end time (ME) are recorded in each single trial. **b** Upper panel: Accuracy of decoding movement direction from the neuronal population activity for the three task conditions. Lower panel: population firing rates. **c** Upper panel: Task-related reduction in trial-to-trial variability ΔFF as a function of trial time; * indicates pairwise significant differences across FF distributions (Wilcoxon signed rank test, two-sided, Conditions 1–2: $p = 0.007$,

Conditions 2–3: $p = 0.03$). Lower panel: Time-resolved estimate of the CV$_2$. A number of total $n = 1215$ samples ($n = 405$ per condition) obtained from $N = 76$ single units entered the analyses in (**b, c**). **d** Architecture of the E/I-clustered attractor network model. Each of the six embedded clusters represents one target direction and can receive excitatory input with the PS and RS stimuli. Each cluster is assigned a direction for which it receives its external stimulus. To match the experimental firing rates in (**b**) we determined 0.1 pA as stimulus amplitude that was kept fixed in all three conditions. The decoder integrates the average cluster firing rates and generates a decision. **e, f** Analysis as in (**b, c**) but for all excitatory neurons in the spiking network model ($n = 7200$ samples per condition). Distributions of FF are pairwise significantly different as determined by a two-sided Wilcoxon signed rank test (Conditions 1–2: $p = 3.6 \times 10^{-13}$, Conditions 2–3: $p = 3.7 \times 10^{-30}$). Decoding accuracy, FF, and CV$_2$ were estimated in a sliding window of 400 ms width.

neuronal dynamics observed in the experimental in vivo data, weak stimuli are required with stimulus amplitudes that lie in the very low range of the stimulus amplitudes considered in Fig. 4.

In summary, the E/I-clustered network robustly captures the reduction in trial-to-trial variability during stimulation due to the reliable switching to the activated state even for low stimulus amplitude retaining a smooth relationship between stimulus strength and firing rate response over the complete stimulus range, while the E-clustered network fails to represent low to moderate population rates in activated clusters and cannot reproduce the reduction in trial-to-trial variability.

### Motor cortical activity reflects target uncertainty during movement preparation and execution

We now analyze monkey behavior and motor cortical single-unit activity during the delayed reach task in its full complexity for monkeys M1 (Fig. 5) and M2 (Fig. S3). As shown in Fig. 5a the monkeys performed the task under three different conditions that varied in the amount of cued target information. At the beginning of the 1 s delay period and in Condition 1, the PS cued a single target by a green light (full target certainty). The target light turned red at the end of the delay period (RS), prompting the monkey to move. In Condition 2, the PS cued two possible targets by two adjacent green lights, one of which was then randomly chosen as the final movement target (single red light) presented as RS. Condition 3 implied the highest target uncertainty during the preparatory period with three possible adjacent targets cued by the PS.

We first analyzed the encoding of movement direction in the motor cortical single-unit activity. To this end, and reconsidering the approach taken by ref. 39, we trained and cross-validated a classifier to predict the direction of the executed movement in each trial based on the neuronal population activity (see Methods). We then computed the decoding accuracy, i.e., the fraction of correctly predicted single-trial movement directions, as a function of trial time as shown in Fig. 5b for M1. In all three task conditions and during the preparatory period the decoding accuracy reaches the theoretical limit that reflects the target information available to the monkey. When full information was available (Condition 1), decoding accuracy approaches unity. When target information was incomplete with either two or three possible targets indicated at PS, the decoding accuracy levels at 1/2 and 1/3, respectively, accurately reflecting target uncertainty. After the RS resolved target ambiguity by indicating the final single movement target, decoding accuracy approached unity in all three conditions.

Next, we asked whether task uncertainty modulates trial-to-trial variability of the single-unit activity. We therefore computed ΔFF as a function of trial time comparing the result across the three different task conditions for monkey M1 in Fig. 5c (cf. Supplementary information for monkey M2). In all three conditions, spike count variability is initially quenched in reaction to the PS. Subsequently, the FF recovers to a different level depending on the experimental condition. In the case of complete target information (Condition 1) the FF remains at a low value, while in Condition 2 it resumes a higher, and in Condition 3 a still higher plateau value that still remains slightly below the baseline level during spontaneous activity. The differences

**Table 2 | Network parameters for the simulation of the monkey task**

| Parameter | Unit | Value |
|---|---|---|
| $N$ | - | 1200(E), 300(I) |
| $E_L$ | mV | 0 |
| $V_{th}$ | mV | 15 |
| $V_R$ | mV | 0 |
| $C_m$ | pF | 1 |
| $\tau_m$ | ms | 20(E), 10(I) |
| $\tau_{syn}$ | ms | 3(E), 2(I) |
| $\tau_r$ | ms | 5 |
| $p_{EE}$ | - | 0.2 |
| $p_{EI}, p_{IE}, p_{II}$ | - | 0.5 |
| $g$ | - | 1.2 |
| $J_{EE}$ | pA | 0.45 |
| $J_{EI}$ | pA | −1.20 |
| $J_{IE}$ | pA | 0.34 |
| $J_{II}$ | pA | −1.83 |
| $I_x$ | pA | $1.25I_{th}$(E), $0.78I_{th}$(I) |
| $Q$ | - | 6 |
| $J_{E+}$ | - | 3.2 |
| $R_J$ | - | 3/4 |

between the task conditions are statistically significant. Thus, the level of trial-to-trial variability of single-neuron spiking activity during the preparatory period directly corresponds to the degree of target ambiguity, with higher uncertainty leading to increased variability and lower uncertainty resulting in reduced variability. Following the RS and during movement execution, the FF shows an additional reduction in Conditions 2 and 3. No additional reduction in FF is visible in the single-target condition. Note that the average single-neuron spiking irregularity ($CV_2$) in Fig. 5c (lower panel) remained constant throughout the task and in all three conditions at a value of $CV_2 \approx 0.8$ that is matched in monkey M2 (Fig. S3c). This rules out that the reduction in FF could be partially caused by an increase in firing regularity due to the moderate increase in average firing rate. As an additional control and to rule out the influence of an increasing firing rate on the dynamic FF reduction, we estimated the mean-matched FF as suggested in[44] obtaining the same results (Fig. S2).

## Motor cortex model provides mechanistic explanation for task-dependent movement encoding and variability dynamics

We now examine whether and how an attractor-based model of the motor cortex can reproduce our experimental observations in the behaving monkey. To this end we propose a functional spiking neural network model that combines the E/I-clustered topology with an additional decoder module to support behavioral decision-making as schematically shown in Fig. 5d. The core of this model consists of six E/I-clusters, one for each target direction. The network parameters (Table 2) were chosen to match the spontaneous experimental firing rates and baseline FF as estimated before PS.

In our behavioral task simulations we mimicked the behavioral monkey experiment by applying the same stimulus protocol. Starting with the onset of the PS (Fig. 5a) we stimulated all excitatory neurons of either one, two, or three clusters with the same constant stimulus current throughout the preparatory period to mimic visual cue information. With the RS, i.e., at the end of the preparatory period, stimulation was maintained for only a single cluster that represented the final target. The optimal stimulus amplitude for replicating the experimental results in M1 was determined as 0.1 pA (0.05 pA in M2, Fig. S3) and thus lies in the very low range of tested amplitudes in Fig. 4.

In a continuous simulation the model was faced with 150 trials in each of the three stimulus conditions.

We subsequently repeated the exact same analyses of directional encoding and of variability dynamics for the spike train recordings during the model simulation as for the experimental in vivo recordings (Fig. 5). The classification analyses, now applied to the entire population of excitatory model neurons, resulted in average decoding accuracies that fully resemble those observed in the experimental data (compare Fig. 5b, e), recovering the optimal decoding scores of 1, 1/2, and 1/3 that reflect the respective target uncertainty in conditions 1, 2, and 3. This can be explained by the probabilistic nature of a switching activation between the stimulated clusters. As an example, Fig. 6a shows the spiking activity of all excitatory neurons in the model during a sample trial of Condition 3. With the onset of the PS a stimulus current was applied to clusters 1, 3, and 4 (counting from the bottom). After cue onset, competition arises between these three clusters. All three clusters now share the same higher probability of being active as compared to the non-stimulated clusters. On average the activation times for all three clusters are thus equal and therefore represent the randomly selected final target with probability 1/3 at each point in time. After RS, only the neurons in cluster 3 that represents the selected final target receive stimulating current input and becomes active.

The time-resolved variability analysis of the model data in Fig. 5c captures the condition-dependent temporal modulation of the average ΔFF observed in vivo for monkey M1 (Fig. 5f) where the reduction in FF is largest for the single target cue and smallest for the triple target cue. The average spike time irregularity ($CV_2$) of the model neurons is essentially constant throughout the trial and independent of the target condition, in full qualitative and quantitative agreement with our experimental results (see Supplementary information and Supplementary Fig. 3 for the model comparison to monkey M2). The reflection of target uncertainty in the trial-to-trial variability during the preparatory period is mechanistically explained by our model. Stimulation of a single cluster (Condition 1) makes it very likely that this E/I-cluster becomes activated, while all other clusters are likely to stay inactive for most of the 1 s preparatory period. Across repeated trials of the same single target cue, neurons of the same stimulated cluster exhibit elevated firing rates while all other neurons are likely to show a low firing rate. Hence, the across-trial variability of the spike count is low. When there is competition between two or three stimulated clusters, the neurons of these clusters share the overall activation time and hence the FF will be higher.

## A simple model of decision-making operating on cluster population activities can explain modulation of behavioral reaction times

Inspired by previous models of perceptual decision-making[54–57], the decoder module in Fig. 5d generates a decision variable associated with each cluster. For each target direction $d$, a leaky integrator governed by an equation of the form

$$\frac{dI_d(t)}{dt} = -\frac{I_d(t)}{\tau_I} + C_d \tag{1}$$

integrates the instantaneous spike count $C_d$ of the corresponding neuron population and forgets with time constant $\tau_I$, representing the integration of target evidence. The decision variable $DV_d(t)$ is formed as

$$DV_d(t) = \frac{I_d(t)}{\sum_{j=1}^{6} I_j(t)} \tag{2}$$

expressing the probability that target direction $d$ represents the correct choice of movement direction at time $t$. This is similar to multi-

**Fig. 6 | Decision model generates task-dependent reaction times in monkey M1.**
**a** Raster plot of excitatory cluster activity for an example trial of Condition 3. Solid curves represent the decision variable (DV; Eq. (2)) associated with each cluster. At $t = 1500$ ms the decoder starts integrating, as indicated by the vertical dashed line. Horizontal dashed lines show the level of the decision threshold $\theta$. To the right, target direction numbers are indicated. During the preparatory period (between PS and RS), clusters 1, 3, and 4 are stimulated. The final target (direction 3) was randomly chosen, resulting in a continued stimulation of only cluster 3 after RS. The black circle indicates the decision for a movement in direction 3 by means of threshold crossing. **b** Histograms of reactions times of the model across behavioral task simulations in all three conditions. **c** Reaction time distributions for monkey M1 across $n = 10,409$, 8448, and 8567 experimental trials in task Condition 1, 2, and 3, respectively.

class classification. A behavioral decision was reached when one of the decision variables crossed the common threshold $\theta$ within a 400 ms interval following the RS. Threshold crossings after that period were counted as unsuccessful trials. This is similar to the monkey experiments where the monkey had to react within a short time limit. If a decision variable was already above the threshold at the beginning of the RS, the decision was counted. The threshold was adjusted such that the average accuracy over all sessions is maximized.

The operation of the decoder is illustrated in Fig. 6a where the respective decision variables $\mathrm{DV}_d(t)$ are superimposed on the neural population activity of each target cluster. In this example trial of Condition 3, the clusters 1, 3 and 4 receive stimulation during the preparatory period while target cluster 3 was randomly chosen as the final target indicated by the RS. At the time of RS onset, cluster 3 is already activated and $\mathrm{DV}_3(t)$ further increases and reaches the threshold with a model reaction time of ~80 ms in this single trial.

Figure 6b, c show the distributions of reaction times produced by the model and experimental data in M1, respectively, for each condition. It can be seen that, as in the experimental data, the average model reaction times in Condition 1 were much shorter than in Conditions 2

and 3. In contrast to the experiment, anticipated responses were not penalized in the model. If the decision variable of the correct direction was already above threshold at RS, the trial was counted as successful. In Condition 1 this was frequently the case. The shape of the reaction time histogram for Condition 1 (dark blue in Fig. 6c) suggests that the monkey displayed a similar behavior. Both the data and the model show on average slightly larger response times in Condition 3 compared to Condition 2. The possibility of having prepared for the wrong direction in the model explains the difference in reaction times between the full information and the ambiguous conditions. The same model was successfully adjusted to account for the behavioral data in M2 with a notably different behavioral strategy (see Supplementary information and Discussion).

## Discussion

Our aim was to construct a functional spiking neural network model for behavioral task performance that can claim biological realism at the level of single neuron spiking statistics within and across trials, and with respect to the generation of realistic behavioral reaction times. We propose a mechanistic model for attractor computation that

extends previous spiking attractor network models featuring excitatory clustering[12,16,17,21–23] by adding clustered inhibitory connectivity motivated by accumulating experimental[27,28,30–33] and theoretical[33,37] evidence for local inhibitory clustering in the neocortex. We show that the proposed architecture ensures local balancing of excitatory and inhibitory synaptic input to individual neurons and supports metastability robustly across a wide range of network parameters, enabling winnerless competition as a signature of cortical attractor computation. The model replicates both, single neuron spiking statistics and behavioral reaction times as measured in two monkeys performing a delayed reaching task in which the degree of prior information about the future movement target is varied. At the neuronal level, we could show that our model can accurately reproduce realistic moderate firing rates, constant spiking irregularity, and the task-related dynamics of trial-to-trial spike count variability. In combination with a simple model for evidence integration and decision-making, our model can explain the variability of behavioral reaction times in both monkeys. The model thus mechanistically links spiking attractor dynamics to context-dependent modulation of neuronal and behavioral variability, extending on the reaction time model by Mazzucato and colleagues[16]. Taken together, our findings suggest joint excitatory and inhibitory clustering as a biological realistic, and powerful mechanism for decision-related computation in the neocortex. We discuss the relevance and limitations of our attractor model and describe future steps of improvement and integration with previously studied models.

Previous studies have considered purely excitatory clustering with a "blanket of inhibition" to model decision-related activity in the cortex. These have been successful in explaining the trial-to-trial variability observed in in vivo recordings[12,16,21–23]. However, in this model architecture metastability can be achieved only in a narrow parameter regime and for small cluster sizes (Fig. 2), which strongly limits its use for functional networks that require robustness against changes in network and input parameters, e.g., due to homeostatic regulation. During cluster activation, single neurons exhibit excessive spike rates[37] (Fig. 4) and an unrealistic regular spiking pattern.

By analyzing the stable fixed points of the mean-field equations for binary networks with purely excitatory clustering, ref. 37 could show that in E-clustered networks switching between attractor states is hampered by the high firing rates attained in active clusters that exhibit a strong self-exciting component. Our proposed remedy of assigning an inhibitory population to each cluster by increasing the corresponding $E \rightarrow I$, $I \rightarrow E$, and $I \rightarrow I$ synaptic strengths provides a robust solution to both problems. Since inhibition is now equally selective, each excitatory population is held in check by its inhibitory counterpart and hence the fixed points of the active and passive clusters move closer together. Due to the balanced excitatory and inhibitory input drive, individual neurons remain in the fluctuation-driven regime across inactive and active cluster states (Fig. 3). This maintains continuous irregular spiking (Fig. 1) and facilitates switching dynamics (Figs. 2, 3). The proposed E+I network topology of ref. 21 extends the E-clustered networks by assigning an inhibitory cluster to each excitatory cluster and increasing the unidirectional $E \rightarrow I$ synaptic strengths. While this increases correlations between these populations it lacks the effect of balancing the input to either of the populations. In line with our analysis of the E-clustered network, the E+I topology of Litwin-Kumar et al. lacks the maintenance of the fluctuation-driven regime during all network states, resulting in the same narrow parameter regime (Fig. 3 and Supplementary Fig. 1). We have shown that our model can depict different system behaviors observable in attractor networks. Depending on the clustering strength $J_{E+}$ the network can show no attractor dynamics, winnerless competition or winner-takes-all dynamics (Fig. 2c) while being in balance and thus reproducing realistic trial-to-trial as well as inter-spike interval variability. Besides the clustering strength, also the stimulus strength can constrain the winnerless competition to a single cluster and thus lead

to an effective winner-takes-all scenario. This covers the functionality of previous work on attractor networks for decision-making and proposes a spiking neuronal implementation that reproduces cortical variability. We conclude that the E/I-clustered topology proposed here provides a robust and biologically realistic network architecture for attractor computation in spiking networks.

Single neuron activity within the neocortex exhibits variability across both short and long-time scales. On a short time scale, this variability is evident in the irregular spacing of inter-spike intervals, while on a long time scale, it manifests as variability in the number of action potentials across repeated experimental trials. Thus, any biologically realistic model of cortical motor control must offer a mechanistic explanation for both forms of variability observed in vivo. Our time-resolved analysis of in vivo single unit activity in the motor cortex shows that spiking irregularity is constant throughout spontaneous activity and behavioral task performance with average values of $CV_2 \approx 0.8$ in both monkeys (Fig. 1 and Supplementary Fig. 3). This is in excellent quantitative agreement with a previous report on the $CV_2$ estimated in three behaving monkeys during an instructed delay period in the motor cortex[45]. While spiking irregularity is generally slightly more regular in motor areas compared to visual and prefrontal areas, the average $CV_2$ across all areas remains below unity[58]. A series of in vitro experiments in rat neocortical slice preparations using somatic noise current injections under controlled stationary input conditions demonstrated the importance of balancing excitatory with inhibitory synaptic input to achieve realistic in vivo-like spiking statistics[41,59,60]. Notably, only a high ratio of inhibition to excitation ($R_i = 0.9$) was able to achieve a high coefficient of variation (CV) of ~0.7, which comes close to our in vivo estimates. Additionally, introducing correlations in synaptic input, a phenomenon commonly observed in spiking neural network simulations, could further enhance spiking irregularity[59,61].

In our E/I-clustered network, excitatory synaptic input is fully balanced with inhibitory synaptic input, while all neurons receive an additional small constant current input. By construction, the balance of synaptic input is maintained during inactivated and activated cluster states (Fig. 3). In our behavioral task simulations, the $CV_2$ thus remained constant in time, independent of firing rate modulations, and throughout spontaneous activity and task-dependent network stimulation with an average $CV_2 \approx 0.8$, closely matching the observed in vivo situation (Figs. 5 and S3). For a stationary spiking process, theoretical consideration predicts that the FF assumes a similar value as the squared coefficient of variation with $FF \approx CV_2^2$ (see Methods). This has been confirmed in in vitro experiments that used noise current injection under controlled stationary input conditions[41,59]. The FF typically even assumes values smaller than $CV_2^2$ due to weak negative serial interval correlations[62], again in full accordance with the theoretical prediction (Eq. (18)). In stark contrast, our in vivo analyses show that trial-by-trial variability is high with $FF \approx 1.8$ during spontaneous activity (i.e., before TS) and thus about three times higher than theoretically expected from the observed spiking irregularity ($FF \approx CV_2^2 \approx 0.65$). Moreover, the FF is modulated during trial time and in a task-related manner while the $CV_2$ stays constant. This discrepancy between in vivo observation and theoretical prediction has led to the interpretation that the observed single unit spiking processes are not stationary across trials, violating the theoretical assumption[45,63]. In the spiking attractor model, individual clusters can randomly switch between an inactivated and an activated state. Consequently, each neuron belonging to a cluster generates less or more output spikes. Across repeated experimental trial observation, the number of generated output spikes is thus highly variable. In our E/I-clustered network, a state of increased (decreased) firing rate is the consequence of increased (decreased) fluctuations of the synaptic input current and membrane potential (fluctuation-driven regime) while the balance of excitation and inhibition is retained and, thus, spiking irregularity remains constant.

The E/I-clustered model not only accounts for realistic neuronal variability dynamics but also reproduces functional and behavioral aspects of movement preparation and decision-making: task-related encoding, and variable reaction times matching the experimental observations. In our experimental design, when the monkey has incomplete information during the preparatory period it can only resolve the ambiguity of multiple targets when the RS signal provides full information. This increased reaction times in M1. We observe the same phenomenon in our model, when adjusting the stimulus amplitude to fit neuronal activity in M1. The mechanistic explanation is that in the multiple-target conditions, the clusters compete and the activity switches between them during the preparation period, which aligns with the previous interpretation of a similar experimental tasks[64,65]. When the RS signal resolves the ambiguity, only one of the clusters retains input stimulation. If this cluster has been active at this point in time, the decision variable reaches the threshold faster (short reaction times) than in the case where a switch is required, leading to longer reaction times (Fig. 6). The same mechanistic explanation underlies short and long reaction times in a recent attractor network model describing behavioral reaction times in anticipatory versus unexpected cues depending on pre-stimulus cluster activation[16]. Kaufmann et al.[65] claim that changes of mind are frequent in decision-making tasks. They show that neuronal activity differs between forced and free changes of mind. They find increased reaction times in trials with suddenly forced decisions, which require re-planning. Our attractor model can provide a mechanistic explanation for their observation as a forced choice requires switching between attractor states and thus additional time for reaching a decision threshold.

In discussing the differences in monkey behavior, our study aligns with the notion that visual stimuli influence the motor cortex when they are actively processed and relevant to task performance[66] with typical response latencies of 150–200 ms[39,67,68]. We posit that attention plays a crucial role in directing this processing, affecting the motor representation accordingly. Notably, individual differences in attentional strategies were evident between monkeys M1 and M2 as observed during training and experimental data collection. Monkey M1 tended to capitalize on the early target information in Condition 1, exhibiting faster reactions and anticipatory movements (Fig. 6c), consistent with the model's behavior. In contrast, Monkey M2 adopted a distinct strategy, disregarding the target cue (PS) and relying solely on the RS for movement initiation across all conditions (Supplementary Fig. 3d). This suggests potentially reduced attentiveness to the target cue in M2, possibly opting for a simpler strategy of waiting for the definitive RS[69]. Intriguingly, the same model with identical network parameters could replicate this behavioral variation by adjusting the stimulus amplitude during the delay period for M2 compared to M1. In summary, our E/I-clustered model not only accurately replicates the behavioral data from both monkeys but also demonstrates remarkable adaptability by capturing individual differences in behavior through only a single parameter adjustment.

Our experimental task involves the perception and processing of visual cues with their uncertainty, the following action selection, movement preparation, and movement execution. Our work here focuses on the action-selection process. In our experiment the monkeys make unrestricted movements, and their hand trajectories were not recorded. Our behavioral read-out is thus restricted to press and release of the buttons. We implemented a simple drift-diffusion decision model[54,70], which allowed us to quantitatively match average behavioral reaction times and to qualitatively match reaction time distributions in the monkey experiments (Fig. 6). We interpret the temporal integration of the population activity (Eq. (1)) as the integration of evidence about a target-specific action. If the decision variable (Eq. (2)) reaches the threshold, the appropriate action is selected and movement initiation is triggered. Our proposed network topology can not produce muscle-control signals and is thus fundamentally different in its aim from other studies that focused on motor preparation and execution. The general class of recurrent neural networks[71–74] and the more constrained case of a balanced networks[75,76] are capable of producing realistic muscle-control signals, which can be used to control kinetic arm models. These results are based on the viewpoint of the motor cortex as a dynamical system. Within these models an initial state is reached due to a specific stimulation followed by a quasi-deterministic relaxation. The muscle-control signal is generated by linear combination of the generated network activity. In these models it remains unclear how and where the decision for a specific movement is formed and how the preparatory signal is generated. Thus, these approaches are complementary to our model and raise future challenges of combining the here presented spiking approach to action selection with models of motor preparation and execution (cf. Supplementary information).

We found good agreement between our model and the spiking statistics of motor cortical neurons regarding the slower component of task-related changes in trial-by-trial variability during the preparatory period. However, in monkey M1 we observe an initial fast and transient reduction of the FF immediately after PS where the FF assumes a minimum before returning to a constant level (Fig. 5c) that is not captured by the attractor model (Fig. 5f). We hypothesize that this fast transient reduction may be caused by spike frequency adaptation[77] (SFA), which acts as a cell-autonomous mechanism of action-potential induced self-inhibition. SFA is ubiquitous in spiking neurons in vertebrate and invertebrate nervous systems and has been evidenced in neocortical neurons[78–81]. Previous theoretical results demonstrated that SFA regularizes the spiking process and hence reduces trial-to-trial variability. Under stationary input conditions SFA introduces negative serial correlations of inter-spike intervals[82] as observed in in vivo and in vitro intracellular recordings from cortical neurons under controlled stationary rate conditions[62,83]. In response to an excitatory stimulus, neurons with SFA in spiking simulations of the random balanced network showed a strong and transient reduction of the FF that relaxes with the time-scale of the SFA-inducing outward conductance[84]. In the next step, we plan to introduce SFA to our E/I-clustered model to investigate whether this can additionally explain the observed fast FF dynamics in response to the visual cue stimulus. The fact that the transient effect of SFA on trial-by-trial variability scales with stimulus amplitude[84] may explain why this was not observed in our data recorded from monkey M2 where a lower stimulus current amplitude was required to match the attractor model to the experimental results (Fig. S3).

Our experimental data is recorded after an extensive training period where both monkeys had learned to accurately perform the task with only very few error trials. In our model we, therefore, assumed that the network connectivity has reached a fixed structure. Future work may investigate how the E/I-clustered network could be learned during training of a task while retaining the local balance of excitation and inhibition[35], for instance through spike-timing-dependent functional plasticity combined with selective stimulation[85–90]. To form and recall these clusters in a stable manner over a long time some form of homeostatic mechanism is crucial. Zenke et al.[87] shows that multiple timescales of homeostatic regulation are necessary to form robust and stable clusters that are functionally relevant. Litwin-Kumar et al.[88] investigate homeostatic synaptic mechanisms that act on $I \to E$ synapses in combination with an $E \to E$ plasticity rule to form clusters that reflect previously experienced stimuli. The inhibitory plasticity in these studies is globally modulated while excitatory neurons form local clusters and are responsible for functional representations. Models that implement structural plasticity in spiking networks provide an alternative mechanism that could support cluster formation[91,92]. Motor cortical pyramidal neurons typically display a rather broad tuning to movement direction[39,93]. In our model, we implemented one cluster per movement target. Each

excitatory neuron belongs to a single cluster that receives target-specific input and, thus, individual neurons cannot exhibit a broad directional tuning. To obtain a more realistic directional tuning behavior in single neurons we propose to implement a modified network topology where clusters that represent neighboring targets share stronger mutual connections to match the target topography in the behavioral space[94]. Alternatively or in addition, target specific network input may be more broadly tuned, stimulating, to a different degree, more than one single cluster.

At any point in time, the population activity of neurons in the motor cortex can be described as a vector in the space that is spanned by the individual neurons, and the evolution of this vector can be studied during the trial. Movement preparation following the perception of the movement instructing cue has been linked to the convergence of the neuronal population activity vector to an optimal subspace[43] such that in each single trial an initial condition will be met that is suitable for movement initiation and execution[95,96]. This theory implies that the trial-by-trial variability reduces from a high spontaneous level where neurons largely act independently to a reduced level when the optimal subspace condition is met[43]. This is in line with our observations. The average population firing rate during the fixed delay preparatory period is similar for movements to different directions and in the three different experimental conditions while the actual target directions can still be decoded accurately[39]. In monkey M1 trial-by-trial variability is reduced to a level that reflects the cued target information during the preparatory period (Fig. 5). In experimental Condition 1 where full target information is cued, the neuronal population activity can reach an initial condition that allows to initiate and execute the appropriate movement, which results in faster anticipatory reaction times. In our attractor model, this is achieved by the activation of a single cluster (Condition 1) and implies that the total neuron population approaches an optimal subspace characterized by the preferential activation of neurons in the target-specific cluster that leads to fast reaction times. In conditions 2 and 3, two or three clusters compete, implying that the dimensionality of the neuronal population activity reduces during the preparatory period, but the optimal subspace that sets the condition for movement initiation and execution can only be reached with full target information, leading to longer reaction times. Monkey M2 displays a different behavioral strategy. It did not perform anticipatory movements in Condition 1 and showed highly similar reaction time distributions in all three conditions. During the preparatory period, the FF shows the same temporal evolution in all three conditions, implying the same slow reduction of dimensionality. In our interpretation, M2 performed movement preparation only after RS and invariably for all three experimental conditions, and thus the optimal subspace will be occupied only after RS. Future studies of attractor network models should aim at a comparative study of the task-dependent evolution of the neuronal activity space between experimental and simulated spiking data in simultaneous recordings from a large population of neurons.

## Methods

### Experiment

**Behavioral task and recordings in the monkey.** The monkey experiments were conducted in Alexa Riehle's lab at the CNRS Marseille, France, in accordance to European and French government regulations. Two monkeys, one male aged 3–4 years (M1) and one female aged 4−5 years (M2) performed a delayed center-out reaching task, which involved three different task conditions that differed in the amount of initial target information available to the monkey as illustrated in Fig. 5a[38,39]. The monkey was seated in front of a panel featuring a hexagonal array of touch-sensitive target LEDs and a central LED indicating the starting position. The monkey initiated a trial by touching the central as LED (trial start, TS). During the 1 s delay period starting at $t = 500$ ms the (PS) provided either complete or incomplete

information about the final movement target and consisted of either a single target LED (Condition 1), two adjacent target LEDs (Condition 2), or three adjacent target LEDs (Condition 3) that lit up in green. At $t = 1500$ ms the (RS) appeared and one of the green target LEDs turned red. This indicated the final movement target and prompted the monkey to move his hand to that target. In Conditions 2 and 3 the final target was randomly chosen among the PS-cued targets, while the other target LEDs went dark. The times of movement onset (MO) and movement end (ME) were recorded and if the monkey touched the correct target LED, the trial was registered as successful and a drop of juice was given as a reward. A premature onset of the behavior before RS led to the abortion of the trial and no reward was given. Only successful trials were analyzed in the present study.

The task conditions of one, two or three possible targets presented during the 1 s preparatory period were executed in blocks and the order of these blocks were randomized across recordings sessions. In each block, 150 trials with randomized target directions were carried out so that each of the directions appeared on average 25 times per condition. Note that to obtain the same number of possible trial types in all conditions, not all possible combinations of directions for the preparatory stimulus were used in Conditions 2 and 3. Since six combinations are possible for Condition 1, only the pairs 1-2, 3-4, and 5-6 were used in Condition 2 and for Condition 3, only two cases occurred (6-1-2, 3-4-5).

Extracellular recordings were obtained with a multielectrode microdrive (Reitböck system; Thomas Recording) to insert transdurally seven independently movable electrodes within the area of the recording chamber, which was positioned over the motor cortex close to the central sulcus in monkey M1 and slightly more anterior in monkey M2 covering in parts primary and dorsal premotor cortex[39]. Online spike sorting resulted in up to seven simultaneously recorded single-unit spike trains[38]. We analyzed a total of 111 (M1) and 110 (M2) single units. Not on all experimental days all three conditions could be successfully completed, resulting in $N = 76$ (M1) and $N = 62$ (M2) single units that were recorded in all three conditions.

### Model

**Spiking network model.** Our spiking network model is composed of leaky integrate-and-fire neurons with exponential synaptic currents where the sub-threshold evolution of the membrane potential $V$ is described by the differential equation

$$\frac{dV}{dt} = \frac{-(V - E_L)}{\tau_m} + \frac{I_{syn} + I_x}{C_m}. \tag{3}$$

In the absence of input, the membrane potential decays exponentially to the resting potential $E_L$ with time constant $\tau_m$. The current $I_{syn}$ represents the synaptic input, $I_x$ is an externally injected current and $C_m$ is the membrane capacitance. If the potential reaches the threshold $V_{th}$ a spike is emitted and $V$ is clamped to a reset voltage $V_r$ for an absolute refractory period $\tau_r$. The synaptic current to a neuron $i$ evolves according to

$$\tau_{syn} \frac{dI_{syn}^i}{dt} = -I_{syn}^i + \sum_j J_{ij} \sum_k \delta\left(t - t_k^j\right) \tag{4}$$

where $t_k^j$ is the time of the arrival of the $k$th spike from presynaptic neuron $j$ and $\delta$ is the Dirac delta function.

To facilitate comparison with previous studies that investigated excitatory cluster topologies we here use similar parameters as provided in ref. 21 and ref. 23 (see Table 1). We briefly explain how we derived the main parameters in the following.

**Calibration of the balanced state.** We follow the same approach as in ref. 37 for the binary networks by requiring that $\sqrt{K}$ excitatory action

potentials arriving within a short time suffice to drive the membrane potential from $E_L$ to $V_{th}$ and hence elicit a spike. For that purpose, we need to compute the deflection in the membrane potential caused by a presynaptic spike.

According to Eq. (4), a spike arriving at $t = 0$ leads to a post-synaptic current of the form

$$I_{psc}(t) = J e^{-t/\tau_{syn}} \Theta(t) \tag{5}$$

where $J$ and $\Theta$ are the synaptic efficacy and step function, respectively. Inserting this into Eq. (3) and integrating with $V = 0$ at $t = 0$ the post-synaptic potential is obtained:

$$PSP(t) = J \frac{\tau_m \tau_{syn}}{\tau_m - \tau_{syn}} \left( e^{-t/\tau_m} - e^{-t/\tau_{syn}} \right) \Theta(t). \tag{6}$$

The maximal deflection $PSP_{max}$ occurs at $t = \frac{\log(\frac{\tau_{syn}}{\tau_m})}{(1/\tau_m - 1/\tau_{syn})}$. Note that the PSP amplitude depends on the synaptic as well as the membrane time constants and is therefore different for each synapse type ($PSP_{max}^{EE}$, $PSP_{max}^{EI}$,...). The scale-free weights are then constructed in the same way as for the binary networks (Eqs. 3–8 in ref. [37]) but weighted by the respective PSP amplitudes:

$$j_{EE} = \frac{V_{th} - E_L}{\sqrt{p_{EE} n_E}} \frac{1}{PSP_{max}^{EE}} \tag{7}$$

$$j_{EI} = -g j_{EE} \frac{p_{EE} n_E}{p_{EI} n_I} \frac{PSP_{max}^{EE}}{PSP_{max}^{EI}} \tag{8}$$

$$j_{IE} = \frac{V_{th} - E_L}{\sqrt{p_{IE} n_E}} \frac{1}{PSP_{max}^{IE}} \tag{9}$$

$$j_{II} = -j_{IE} \frac{p_{IE} n_E}{p_{II} n_I} \frac{PSP_{max}^{IE}}{PSP_{max}^{II}} \tag{10}$$

where $g$ is the relative strength of inhibition. The final weights $J_{\alpha\beta}$ are obtained by dividing by $\sqrt{N}$.

Since we are interested in the temporal dynamics of neuronal variability, we modeled external inputs as constant currents to ensure that all variability arises deterministically inside the network rather than stemming from externally generated Poisson input. In analogy to the "threshold rate" of Brunel[15], the external current $I_x$ is expressed in terms of the current required to reach the threshold in the absence of synaptic input:

$$I_{th} = \frac{V_{th} - E_L}{\tau_m} C_m. \tag{11}$$

A complex interplay exists between the $E$ and $I$ firing rates and the magnitude of the external currents to the populations. The tuning of the injected currents required to obtain the desired firing rates of 3 and 5 spikes per second for the $E$ and $I$ populations, respectively, was therefore achieved by modeling single units with Poissonian inputs mimicking the network input at the target firing rates. The external inputs could then be increased until the modeled units fired on average at the required rates.

Before introducing structured connectivity we first ensured that the network configuration was operating in the (AI) regime. Irregularity was measured using the (CV$^2$) (as explained in the Data analysis section). Synchrony of measures such as the instantaneous firing rate

or the membrane potential in neural networks can be quantified as[97]:

$$\chi = \sqrt{\frac{\sigma_{pop}^2}{\langle \sigma_i^2 \rangle}}. \tag{12}$$

Here $\sigma_{pop}^2$ is the variance of the the population average and $\langle \sigma_i^2 \rangle$ is the average over the individual units' variances. The measure gives unity for totally synchronized activity and for asynchronous activity in networks of size $N$, one expects $\chi \sim \mathcal{O}\left(\frac{1}{\sqrt{N}}\right)$. Since recording all membrane potentials in simulations is computationally expensive, we computed $\chi$ on spike counts measured in bins of 20 ms.

It can be seen in Supplementary Fig. 4 that the networks show the usual characteristics of the balanced state. When excitation dominates, synchronous-regular firing near the saturation rate $1/\tau_r$ is observed. The (AI) state occurs when $g$ is sufficiently large for inhibition to dominate (Supplementary Fig. 4a). As in the binary network[37], we choose $g = 1.2$, where $\chi = 0.02 \sim 1/\sqrt{N}$ and CV$^2 = 0.73$ (Supplementary Fig. 4b). The raster plot shows no discernible structure (Supplementary Fig. 4c) and the average firing rate is low and constant over time (Supplementary Fig. 4d). The synaptic currents from excitatory and inhibitory inputs and the external current $I_x$ cancel so that the net input fluctuates around zero (Supplementary Fig. 4e). Hence, the membrane potentials fluctuate at low values and only occasionally reach the threshold to produce a spike (Supplementary Fig. 4f). The parameters used for all simulations in this section are summarized in Table 1.

**E and E/I-clustered networks.** We follow the same connectivity scheme that we introduced for binary networks in our previous work[37]. In summary, for the E-clustered networks, we first divide the excitatory population into $Q$ equally sized clusters with uniform connection probability. Subsequently, we potentiate the synaptic connection within each cluster by a factor $J_{E+}$, referred to as cluster strength. A value of $J_{E+} = 1$ represents the random balanced network and the larger $J_{E+}$, the stronger the weights within the formed clusters. To maintain the overall balance, we decrease the weights among units belonging to different clusters by a factor

$$J_{\alpha-} = \frac{Q - J_{\alpha+}}{Q - 1}, \tag{13}$$

where $\alpha = E$.

For the E/I-clustered networks, we divide not only the excitatory population but also the inhibitory population into $Q$ clusters. We introduce distinct excitatory and inhibitory clustering strengths, $J_{E+}$ and $J_{I+}$. Then we require that each excitatory cluster selectively influences its corresponding inhibitory cluster and vice versa by increasing the corresponding EI, IE and II weights by a factor $J_{I+}$. Balance is again maintained by rescaling across-cluster weights according to Eq. (13) for $\alpha = I$. The overall scaling within and across clusters (indicated by the superscripts in/out) is expressed as follows: $J_{EE}^{in} = J_{E+} J_{EE}$, $J_{EE}^{out} = J_{E-} J_{EE}$, $J_{\alpha\beta}^{in} = J_{I+} J_{\alpha\beta}$, $J_{\alpha\beta}^{out} = J_{I-} J_{\alpha\beta}$ for $\alpha\beta \in (EI, IE, II)$. We have shown in ref. [37] that the inhibitory clustering needs to be weaker than the excitatory clustering to obtain realistic firing rates. To capture this relationship, we introduce a proportionality factor $R_J$, such that

$$J_{I+} = 1 + R_J(J_{E+} - 1), \tag{14}$$

where $0 \leq R_J \leq 1$. A value of $R_J = 1$ implies the same cluster strength for inhibitory and excitatory clusters ($J_{E+} = J_{I+}$) and $R_J = 0$ makes the inhibitory population unclustered ($J_{I+} = 1$, which represents the E-clustered networks). Throughout the current study, we use $R_J = 3/4$ based on our previous results[37] where we showed that this value of $R_J$ can prevent firing rate saturation in up states.

**Attractor model of motor cortex for simulation of the behavioral monkey task.** We designed a model with six E/I-clusters with 200/50 excitatory/inhibitory neurons each. We adjusted the clustering parameter $J_{E+}$ for the smaller network size to achieve robust metastability under spontaneous network conditions and an average FF that approximates the average experimental value. The external input currents were slightly adapted to obtain spontaneous firing rates of the excitatory neurons matching the average experimental value. The inhibitory neurons were tuned to have slightly higher firing rates. To match the task evoked firing rates and FF the only free parameter is the stimulus amplitude. No additional parameter tuning was performed and all model parameters are listed in Table 2.

In an ongoing simulation we define successive trials of 2 s length. We randomly define the start of the first trial (TS). This is followed after 500 ms by the onset of the PS and after another 1000 ms by the onset of the RS, which lasted for 400 ms. PS and RS were realized as constant stimulating currents $I_{stim}$. After each trial we draw a random inter-trial interval in the range of 1.5–1.7 s before we start the next trial, to allow the network to relax to its spontaneous state. The variance in this relaxation period was intended to avoid any effects of periodicity. For each task condition the model executed 150 trials and for analysis the trials were cut from this long continuous simulation. All analyses of model spiking data were performed identically to the analyses of the in vivo spiking data. Our setup implies that each neuron in our model is sharply selective to only a single target direction.

The threshold $\theta$ applied to the decision variable DV (Eq. (2)) was adjusted to maximize the performance of the model. The time constant of integration $\tau_I$ was set to 50 ms which represents an intermediate value between very fast reactions directly when the threshold is reached at RS and very slow integration where the threshold was not reached during the RS interval.

### Data analysis

**Quantifying neural variability.** The FF quantifies the dispersion of spike counts for single neurons across repeated observations of the same experimental condition (trials)

$$FF = \frac{\sigma_c^2}{\mu_c},$$ (15)

where $\sigma_c^2$ and $\mu_c$ are the variance and mean count over trials. The estimation of the FF is biased towards unity for small estimation windows. However, this bias quickly becomes negligible when the estimation window is several multiples of the mean (ISI)[41,63].

Interval statistics are usually characterized by the (CV) of the empirical(ISI) distribution,

$$CV = \frac{\sigma_{ISI}}{\mu_{ISI}}.$$ (16)

Here, $\sigma_{ISI}$ and $\mu_{ISI}$ are the standard deviation and mean of the intervals between action potentials. Estimating the cv requires some caution, as modulations in firing rate increase the interval variability. Another problem with estimating the (CV) follows from finite-size estimation windows. In an estimation window of width $T$, only ISIs < $T$ can be observed. If the underlying process has non-zero probabilities for larger intervals, the (CV) will be underestimated; this effect is known as right-censoring[41,98]. For estimating the interval variability we therefore used the $CV_2$[40,42], which is computed for all pairs of consecutive intervals as:

$$CV_2 = 2\left\langle \frac{|\tau - \tau'|}{\tau + \tau'} \right\rangle.$$ (17)

Here, $\langle...\rangle$ denotes averaging and $\tau$ and $\tau'$ are consecutive (ISI). The $CV_2$ is assumed to be largely unaffected by firing rate changes on time scales that are slower than two consecutive intervals.

A neuron generates action potentials and a spike train may be considered a realization of a stochastic point process[99]. We may thus interpret interval and count variability in the light of stochastic point process theory. For any defined point process the expectation values for the FF and (CV) can be computed and are related. For stationary processes, i.e., when the point process definition is fix and the point process intensity (i.e., the firing rate in neuronal terms) is constant, we obtain[62,63,100,101]

$$\lim_{T \to \infty} FF = CV_\infty^2 (1 + 2\xi) \qquad \text{with} \quad \xi = \sum_{i=1}^{\infty} \xi_i$$ (18)

where $\xi$ denotes the ith-order linear correlation coefficient for pairs of intervals ($ISI_k$, $ISI_{k+i}$). In practical terms, the limit notation reflects the estimation biases (see above) for finite observation windows of length $T$. In the case of a renewal process where the stochastic point process is defined by the inter-spike interval distribution and, thus, no serial correlations of inter-spike intervals exist, we expect FF ≈ $CV^2$ for long $T$, for the special case of the Poisson process this simplifies to FF = $CV^2$ = 1 irrespective of the length $T$ of the observation interval.

In the present study we consider an experimental trial design where in each trial the monkeys (and our models) perform a given behavioral task. For our analyses, we grouped trials that belong to the same experimental condition and with identical preparatory target cue (PS) and final target cue (RS) presentation. For each single unit we then count the number of spikes in each trial and within the defined observation window to compute the FF across a group of trials. For the same observation window we computed the per-trial average $CV_2$ before averaging across trials. Using a sliding observation window allows to observe the task-related temporal dynamics of the average FF and $CV_2$. To allow a fair comparison of variability statistics across conditions, additional precautions were taken. We required that units had at least ten spikes in the 2 s interval after trial start and that at least ten trials were recorded per condition and cued target direction. To enable the comparison across experimental conditions, we only included units and cued target directions where those criteria were met in all conditions. Under the assumption of stationarity we expect the relation given in Eq. (18) for FF and $CV_2$. Strong deviations from this relation indicate non-stationarity across trials[41,63].

**Decoding of movement direction.** To assess how much directional information is contained in the population activity we reproduced the approach of ref. 39 and constructed pseudo-populations of all available units as follows: the data set was divided into five groups and each group in turn served as the test set while the model was trained on the remaining groups, and the average score of the five models was calculated. At each point in time and using five-fold cross-validation we computed the decoding accuracy of the logistic regression classifier as the fraction of correctly predicted movement directions averaged over all six different directions as

$$\text{Decoding accuracy} = \frac{1}{C} \sum_{c=1}^{C} \frac{N_{correct}^c}{N_{total}^c}.$$ (19)

### Reporting summary

Further information on research design is available in the Nature Portfolio Reporting Summary linked to this article.

## Data availability

All experimental data in this study have been deposited at the German Neuroinformatics Node (G-Node) under accession code https://gin.g-node.org/nawrotlab/delayed_center-out_uncertainty_Riehle, https://

doi.org/10.12751/g-node.rz77m8[102]. Obtained simulation data is available under https://gin.g-node.org/nawrotlab/EI_clustered_network.

## Code availability

All code to reproduce the figures, including simulation code and code to analyze the experimental and simulated data are available under accession code https://github.com/nawrotlab/ClusteredNetwork_pub, https://doi.org/10.5281/zenodo.11353864[103].

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

## Acknowledgements

This project was funded by the German Research Foundation (DFG), in parts through the Collaborative Research Center "Motor Control in Health and Disease" (DFG-SFB 1451, Project A06, ID 431549029 to M.P.N.) and under the Institutional Strategy of the University of Cologne within the German Excellence Initiative (DFG-ZUK 81/1 to MN and SA). S.J.v.A. received funding through European Union's Horizon 2020 Framework Programme for Research and Innovation under grant agreement number 945539 (Human Brain Project SGA3). F.J.S. was funded through the DFG Research Training Group Neural Circuit Analysis (DFG-RTG 1960, grant no. 233886668 to M.P.N.). We thank Dr. Moritz Deger for fruitful discussions and support in the early phase of the project. The freehand drawing of the monkey in Fig. 1a was produced by T.R.

## Author contributions

Conceptualization, V.R., T.R., A.R., and M.P.N.; experimental task design and experiments, A.R.; data analysis, V.R., T.R., F.J.S., and M.P.N.; computational modeling, V.R., T.R., and F.J.S.; writing—original draft, V.R. and M.P.N.; writing—review & editing, V.R., F.J.S., S.J.v.A., A.R., M.P.N., funding acquisition, S.J.v.A. and M.P.N.

## Funding

## Competing interests

The authors declare no competing interests.
