## [Peer Review File · Nature Communications]

Spiking attractor model of motor cortex explains modulation of neural and behavioral variability by prior target informationREVIEWER COMMENTS

Reviewer #1 (Remarks to the Author):

This study by Rostami, Rost et al. addresses the question of whether the neural and behavioral variability observed in monkeys performing a delayed center-out reach task is a signature of attractor computation. To that end, the authors constructed a neural network model with excitatory neuron clusters balanced with inhibitory neuron clusters, which recapitulated the dynamics of trial-to-trial neural variability and spiking irregularity observed in monkey motor cortical neural recordings. This model also provided a neural explanation for the reaction-time variability observed in monkey behavior in the context of incomplete cue information.

The emphasis on adding inhibitory clusters to achieve robust metastability is an important step forward for spiking neural networks to model the biological neural networks more closely. To support the biological plausibility of their neural network model, the authors showed that the neural variability of this model looked similar to monkey neural recordings and that the reaction-time distributions of this model during decision-making resembled the monkey behavior. However, I think the connection between the neural network model and monkey neurophysiology data is not convincingly tight (see below for detailed reasons). Moreover, attractor dynamics have been shown by previous work as a signature neural pattern during decision-making and could explain many behavioral observations (e.g., Finkelstein et al., *Nat Neurosci*, 2021) – it is thus unclear to me how attractor computations shown in this work add new knowledge to our understanding of the neural dynamics underlying decision-making.

Taken together, I think that the work needs major revisions for the scope of the advance presented in the manuscript to be more sufficient. First, the paper will improve if the positive aspects of the work are brought out more impactfully in the analyses and writing (see major comments 1 and 2). Second, there are several critical experimental details missing that could enhance the significance of the E/I-clustered neural network but the amount of work needed is substantial given the animal model being used (see major comments 4 and 5).

Major questions / suggestions:

1. Although the authors compared their E/I-clustered model to an E-clustered model to show the importance of inhibitory clusters, it is unclear how this E/I-clustered network outperforms previous E/I balanced networks: for instance, Hennequin et al. 2014 showed that their E/I-balanced network with constructed inhibitory architecture was able to reproduce arm muscle dynamics. The current study may be more helpful to the field if the authors could explicitly compare this work to other prior E/I-balanced networks that modeled monkey motor cortical data. I'd also suggest that the authors consider using the current model to predict properties of the monkey's actual movements (e.g., variability of movement trajectories) in addition to the reaction-time predictions to further support the idea that this neural network model is biologically plausible.

2. It would help emphasize the importance of the E/I-clustered model if the authors could clarify the motivation for choosing certain metrics to compare the current model with previous models and why those metrics are critical measurements of model performance. For instance, why is the linearity between stimulus strength and neural response important? It is possible that in a biological neural network in the brain, the relationship between stimulus and neural activity is nonlinear, which would weaken the similarity between the E/I-clustered model and biological neural networks. Similarly, why are the trial-to-trial neural variability and spiking irregularity key metrics of neuronal dynamics?

3. To mimic the center-out reach task, the authors embedded 6 clusters in their model. However, based on prior neurophysiology studies, I think that neurons in the monkey motor cortex generating movements toward different targets are not separated into clusters. Instead, the same neural population generates different activity patterns to produce those different movements. Could the authors resolve this discrepancy or explain why the separate clusters provide a reasonable approximation of the neurophysiological observations?

4. In a different type of the delayed center-out task (e.g., Kaufman et al., *eNeuro*, 2016), the

variance of reaction time (RT) is due to different durations of the delay period rather than the uncertainty of targets/incomplete information (i.e., whether there's enough time for the preparatory neural states to evolve to an "optimal" initial state). Could the E/I-clustered model explain the RT variance in this case? It would be very valuable to the field if the authors could use their model to examine the potential neural mechanism underlying reaction-time variance in general.

5. I recognize that it would be tricky to collect new data from a second monkey, but $N=2$ is the typical practice of monkey neurophysiology studies. The current results would be further strengthened if the authors could repeat them in a second monkey, ideally from existing datasets.

Minor suggestions:

1. Figure 1: To help readers understand the plots without going back to the main text, I'd suggest adding the full names of PS and RS to figure 1a legend.

2. Figure 1b,c is trying to replicate the decrease of FF after PS as observed in monkey motor cortical neurons. Since the monkey data only involved one type of visual stimulus, could the authors clarify what the weak and strong stimuli used in the model correspond to in the monkey data?

3. Line 311: "stimulate" should be "stimulated".

4. Figure 6b,c: Could the authors explain why the distributions of reaction time for conditions 2 and 3 are much wider in the model than the monkey data? I'd also suggest adding statistical tests to quantitatively compare the distributions.

5. Does the model capture other behavioral variability in addition to the reaction-time variability, such as the end-point variance of reaches? It is also not clear to me if the replication of the behavioral variability is due to adding the inhibitory clusters or is intrinsic to any network model (i.e., both E-clustered and E/I-clustered models)? Could the authors compare the behavioral predictions of both models?

6. Related to my last comment, the reaction time prediction doesn't seem to be unique to the E/I-clustered network: my understanding is that for any network with competing ramping activity towards multiple decision boundaries, there would be an uncertainty/evidence-dependent variance in reaction time. Could the authors rule out this possibility? Or alternatively, could the authors show the unique strengths of their model in terms of predicting behavior? For instance, as I mentioned in previous comments, maybe the authors could test whether their model is capable of predicting behavioral parameters in addition to reaction-time variance and outperforms other E/I-balanced networks.

Reviewer #2 (Remarks to the Author):

The standard balanced network model produces an irregular spike train observed in vivo. To further explain slow spike rate fluctuations and trial-to-trial variability in the cortex, balanced network models with clustered excitatory connections have been proposed. Yet, these models require fine-tuning. In addition, the spike rate becomes extremely high when trial-to-trial variability is matched to experimental data. Rostami et al implemented and analyzed a balanced network model with excitatory and inhibitory clustering. They have shown the model explains task-related dynamics in the primate motor cortex well, including a decline in trial-to-trial variability across task conditions. These findings are interesting and important, yet multiple issues need to be addressed, as listed below.

Major

1) The authors claim that the EI-clustered network is a novel architecture (P.4 L.69). Yet, the EI-clustered network has been reported in Litwin-Kumar et al. (Fig8) to explain trial-to-trial variability in inhibitory neurons. I agree that analysis in Litwin-Kumar is limited, and authors have reported new features of the network. Yet, this should be clarified in the manuscript.

2) Metastability is not explained well, despite its importance in this paper. The introduction should be expanded for non-experts.

3) Spike rates (SR) need to be analyzed in more detail. It is important to show SR is matched besides FF and CV2 to claim the model captures task-related dynamics. Most figures and analyses do not show SR and PSTH (e.g., Fig1 and 5).

Fig 4 shows that spike rates (SR) need to be extremely high in the E-clustered network to reduce FF, while it is not the case for the E-I network. Do both FF and SR in the E-I network match the data (text only mentions that FF matches the data)?

4) P.9 L.185 "Local balance of excitation and inhibition facilitates attractor dynamics and maintain spiking irregularity". This part of the paper beautifully explains how the E clustered network results in mean driven regime to produce regular spiking. In contrast, the E-I clustered network maintains under a fluctuation-driven regime to keep the irregular spiking. But as far as I understand, it says nothing about "facilitates attractor dynamics". Please remove it or explain what it means.

5) It is critical to rule out whether declining FF is due to rising SR. Show SR and mean matched FF (Figs 1 and 5).

Minor

1) Figure1. It looks odd that FF declines before the task onset. Use a causal sliding window.

2) In Figure 5f, there is no FF decline at the task onset, unlike in the data (5c). Please comment on this.

3) The model in Figs 5D-F & 6 predicts competition between three clusters tuned for different targets in the triple target condition. If you happen to have simultaneously recorded data, do you see evidence of "competition" among neurons with different tuning (i.e., different clusters)?

4) Daie et al., NN, 2021 uncovered a cluster-like structure in the frontal cortex of mice performing a delayed response task. It may be worth citing as it supports the clustered network in a similar task.

Point-to-point response to the reviewers' comments

Reviewer #1 (Remarks to the Author):

This study by Rostami, Rost et al. addresses the question of whether the neural and behavioral variability observed in monkeys performing a delayed center-out reach task is a signature of attractor computation. To that end, the authors constructed a neural network model with excitatory neuron clusters balanced with inhibitory neuron clusters, which recapitulated the dynamics of trial-to-trial neural variability and spiking irregularity observed in monkey motor cortical neural recordings. This model also provided a neural explanation for the reaction-time variability observed in monkey behavior in the context of incomplete cue information.

The emphasis on adding inhibitory clusters to achieve robust metastability is an important step forward for spiking neural networks to model the biological neural networks more closely. To support the biological plausibility of their neural network model, the authors showed that the neural variability of this model looked similar to monkey neural recordings and that the reaction-time distributions of this model during decision-making resembled the monkey behavior. However, I think the connection between the neural network model and monkey neurophysiology data is not convincingly tight (see below for detailed reasons). Moreover, attractor dynamics have been shown by previous work as a signature neural pattern during decision-making and could explain many behavioral observations (e.g., Finkelstein et al., Nat Neurosci, 2021) – it is thus unclear to me how attractor computations shown in this work add new knowledge to our understanding of the neural dynamics underlying decision-making. Taken together, I think that the work needs major revisions for the scope of the advance presented in the manuscript to be more sufficient. First, the paper will improve if the positive aspects of the work are brought out more impactfully in the analyses and writing (see major comments 1 and 2). Second, there are several critical experimental details missing that could enhance the significance of the E/I-clustered neural network but the amount of work needed is substantial given the animal model being used (see major comments 4 and 5).

We thank the reviewer for her/his positive, constructive and insightful comments that helped us to significantly improve the manuscript during our thorough revision. We apologize for the severely delayed revision due to personal reasons.

Below we reply in detail to all major and minor comments and provide additional results accordingly. In particular we have analyzed data from a second monkey that performed the identical task. The results are again well matched by our model simulations (new Supplemental Figure 5).

We substantially revised our Discussion in all parts, accommodating the suggestions of both reviewers; in particular we reference and discuss additional relevant work on attractor dynamics and decision making. We believe that, indeed, our contribution here is novel in several aspects and are confident that we can convince the reviewer in light of our extended results and

improved discussion. Furthermore, we have worked extensively on including our model as a part of the Nest Simulator library to make our model available to the community.

In the revised MS, text changes in Abstract, Introduction, Results and Methods have been marked in blue color with the exception of minor corrections (such as typos). Due to a substantial change and extension of our Discussion we marked this complete section in blue text color without indicating deleted text to ease reading. We have added substantial supplemental material with additional novel analyses and results.

Major questions / suggestions:

1. Although the authors compared their E/I-clustered model to an E-clustered model to show the importance of inhibitory clusters, it is unclear how this E/I-clustered network outperforms previous E/I balanced networks: for instance, Hennequin et al. 2014 showed that their E/I-balanced network with constructed inhibitory architecture was able to reproduce arm muscle dynamics. The current study may be more helpful to the field if the authors could explicitly compare this work to other prior E/I-balanced networks that modeled monkey motor cortical data. I'd also suggest that the authors consider using the current model to predict properties of the monkey's actual movements (e.g., variability of movement trajectories) in addition to the reaction-time predictions to further support the idea that this neural network model is biologically plausible.

i) Performance of E/I-clustered model relative to the random balanced network (RBN)

We believe it is important to match both, neurophysiological measurements with respect to neuronal response and spiking properties, population coding and its relevance for explaining behavioral aspects. The high trial-by-trial variability as measured by the FF in mammalian cortices (FF on the order of 1.5 and higher) cannot be explained or reproduced by the balanced random balanced network (RBN), nor by *in vitro* experiments where stationary noise currents injection cannot drive FF and CV above unity. We have now made sure that, in the Introduction and in the Discussion, the relevant work mentioned above is cited and discussed, see also our reply to major comment 2 below. In light of our monkey data and behavioral task, the RBN cannot reproduce any of our crucial results. In particular it cannot explain the overall high FF and the task-dependent reduction of the FF or its distinct behavior in the three different task conditions. Our study strongly supports existing evidence that attractor dynamics in the cortical network is causal for the overall high Fano factors that do not match the theoretical expectation based on the low spiking irregularity (CV).

ii) Relation of the E/I-clustered attractor network to Hennequin et al.:

We now provide a brief discussion of the model in Hennequin et al. (2014) in the revised Discussion section. As outlined in detail below, the two network types differ fundamentally in their architecture and the aims for establishing and analyzing these models were different. We believe that their functionality is not only different but also complementary, and in our discussion we suggest a combination of both network types where the E/I-clustered - based attractor

network acts as an action selection stage, while the network of Hennequin et al. subserves the formation of realistic muscle control dynamics during execution.

To fully understand the model in Hennequin et al., 2014 we have implemented their rate-based model, reproduced the described dynamics, and successfully mapped this to a 2D movement. For each network instance the connectivity between the 100 excitatory and 100 inhibitory rate neurons is tuned such that the Eigenvalues of the connectivity matrix are stable. A slowly increasing, i.e. ramping, input to the network (deterministically) results in a nearly unchanged population-averaged firing rate but single neurons strongly deviate from their baseline firing rate. Switching this input off results in a (deterministic) relaxation that generates firing rate profiles with rate dynamics that can resemble realistic (slow) arm muscle dynamics and, by linear combination, can be used to produce muscle EMG-like profiles. This is an important step towards the realistic simulation of arm movement trajectories. The authors then resemble their rate network by a spiking network. Based on the connectivity (now 200 E and 200 I rate units) with desired Eigenvalue spectrum, each excitatory rate unit is replaced by a population of 60 excitatory spiking neurons and each inhibitory by a population of 15 inhibitory spiking neurons. 50% of the connectivity in this network is distributed randomly, as in the RBN network. As the authors point out, this is to retain irregular spiking similar to the RBN. The other 50% of connections chosen to match the rate-model derived connectivity. This suffices to largely ensure the subcritical network state.

The differences of their network to the E/I-clustered network suggested in our present study with respect to architecture and functional interpretation are prominent. The architecture is fundamentally different: First, in Hennequin et al. the excitatory and inhibitory populations do not form strong connections among the neurons within a population (our definition of a cluster). Rather, 50% of connections are random across the entire network and the other 50% of connections are formed with postsynaptic neurons outside their own population. Thus, following the excitatory cluster definition in Litwin-Kumar et al. and our definition of E and I clusters here, the populations in Hennequin et al. do not establish clusters. Second, near balance of excitation and inhibition and irregularity of spiking in Hennequin et al. is achieved by the 50% random connectivity across the entire network (similar to the RBN). In our case *locally* balanced input conditions are established by forming pairs of E and I clusters through reciprocal high connection weights. Third, our network follows stochastic connectivity rules but we do not apply any algorithm for deletion or insertion of connections based on the Eigenvalues of the connectivity matrix.

With respect to functionality the two networks also show fundamental differences and, as we believe, are rather of complementary nature. First, the E/I-clustered network architecture here was designed to show the desired feature of metastability in the resting state (i.e. for common constant input to all neurons) to support winnerless competition among E/I-clusters while explaining the neuronal variability dynamics observed *in vivo*. To our understanding of the network in Hennequin et al., this network does not exhibit metastability and attractor dynamics. The spiking irregularity (CV) is too high to match the experimental situation analyzed in the present MS and trial-to-trial variability (FF) is not considered in their paper. On the other hand,

the desired relaxation dynamics that, in relevant combinations, can represent muscle control dynamics (and arm movement kinematics) is a key feature of their design of a subcritical network, while our network was not designed to produce such a dynamics. Finally, the metastability in our network supports representation of multiple target-specific movement representation as competing representations and stochastic switching dynamics to a final target. The metastability and a stimulus-induced increased probability of cluster activation explains the observed neuronal dynamics in terms of firing rates, and (dynamic) trial-to-trial variability and specifically its difference across the three different experimental conditions. The intrinsic stochasticity is also expressed in the across-trial variability of reaction times. In the model of Hennequin et al. the (population) firing rates modulate deterministically during stimulation and relaxation and it will fail to capture a realistic value of trial-to-trial FF (in the motor cortices typically on the order of 1,5-3) during spontaneous activity, nor can it capture the task-related reduction of FF. Reaction times (not considered in Hennequin et al.) will not be variable due to lack of intrinsically stochastic behavior. For us, a major aim, however, was to explain qualitatively and quantitatively the experimentally observed neuronal dynamics (rate, FF, CV) *and* to exploit its functionality for competing and simultaneous target representation.

We believe that matching different signatures of *in vivo* single neuronal and population activity, and reaction time variability within and across the three different experimental conditions is a major achievement and indicates that our model provides a biologically realistic mechanistic and functional description.

iii) Prediction of trajectories

In the present task the monkey is freely reaching with his arm and hand to target positions on a vertical plane. No manipulandum or other means of trajectory measurement has been used to allow for unrestricted movements. We thus only have at hand the respective behavioral events (touch or release of the touch-sensitive LEDs).

As outlined above, we would like to consider a combined approach with the ideas proposed in Hennequin et al, in the future and we would like to apply these to a new currently collected experimental data set with our experimental collaborators to match movement duration and trajectories during an instructed delay task.

2. It would help emphasize the importance of the E/I-clustered model if the authors could clarify the motivation for choosing certain metrics to compare the current model with previous models and why those metrics are critical measurements of model performance. For instance, why is the linearity between stimulus strength and neural response important? It is possible that in a biological neural network in the brain, the relationship between stimulus and neural activity is nonlinear, which would weaken the similarity between the E/I-clustered model and biological neural networks. Similarly, why are the trial-to-trial neural variability and spiking irregularity key metrics of neuronal dynamics?

We thank the reviewer for this helpful comment. In the revised MS we now carefully improved the description and interpretation of the chosen metrics in the Methods and in the Results part,

and we now discuss in detail the relevance and relation of spiking irregularity and trial-to-trial count variability and the interpretation thereof.

Stimulus-response relation.

With our analyses in Fig. 4b we did not want to stress the apparent linearity of the stimulus-response relation in the E/I network and we may have overemphasized this. The important aspect there is, that the E/I network permits activation across a large dynamic range of firing rates that match the biologically realistic regime as observed in our (and others) experimental recordings from motor cortices. Specifically, in the regime of low stimulus amplitudes (Fig. 4b), these can reliably be mapped to a range of realistic firing rates in the activated clusters matching our mean-field prediction (Rost et al., 2018). In the E/I network, the FF is robustly reduced even in this low range of stimulus amplitudes (Fig. 4d). The transfer function for the E network, on the contrary, does not exhibit a smooth input-response function in the lower input regime. Rather, it exhibits an abrupt increase in the cluster firing rates across a narrow stimulus range of approx. 0.05-0.08 pA. This is accompanied by a strong increase in the FF rather than a decrease for stimulus amplitudes up to 0.4 pA, matching the examples provided in Fig. 1b. Our interpretation of rate and Fano factor in E-clustered network is as follows: Low amplitude stimulation leads to a high firing rate state only in some of the trials and not others, resulting in an abrupt jump in average firing rate and increase of the FF. Thus achieving decrease in Fano factor and keeping the firing rate at the realistic regime is not possible with this type of network architecture. On the other hand, the E/I-clustered network model can flexibly produce a rich dynamical regime matching rates, FFs and CV2s observed in experimental data.

In the revised version of our MS we have now made several changes to better explain our results and rationale. For Fig. 4b we now chose a slightly different analysis approach reporting average response firing rate of the activated clusters (that exceed the median rate of the non-stimulated clusters). This emphasizes the abrupt change from essentially zero activation to high average firing rates in the activated clusters observed for the E-clustered network; this also supports interpretation in light of our previous theoretical finding (Rost et al., 2018). We now avoid emphasizing a near-linear relationship for the E/I-clustered network but stress the smooth relation between stimulus amplitude and firing rates across the entire tested stimulus range for the E/I network.

FF and CV are key statistical metrics of neuronal dynamics.

The fact that trial-to-trial variability in the (motor) cortex is high - higher than expected from the interval variability - and undergoes a task-related reduction is important. It indicates that, across trials, the network is observed in different states, which can explain the high FF. At the same time the spiking irregularity is constant throughout spontaneous activity and task performance at a reasonable level that can be explained by (near) balance of excitatory and inhibitory synaptic inputs. To our understanding, a biologically realistic model should be able to mechanistically explain the relation of an excessive and task-modulated FF and a constant CV. Our model provides a mechanistic explanation and quantitative match of both metrics and their dynamics, and the identical network model can fit the data of both monkeys. We now motivate our focus on

these metrics in the Results section and provide a more theoretical account in the Methods section. In the Discussion we explicitly discuss the importance of these metrics and their relation to the *in vivo* data and our suggested model.

3. To mimic the center-out reach task, the authors embedded 6 clusters in their model. However, based on prior neurophysiology studies, I think that neurons in the monkey motor cortex generating movements toward different targets are not separated into clusters. Instead, the same neural population generates different activity patterns to produce those different movements. Could the authors resolve this discrepancy or explain why the separate clusters provide a reasonable approximation of the neurophysiological observations?

Indeed, in our model and due to the separated clusters, individual model neurons are specifically tuned to a single target direction. While this is a current limitation of our model, we believe this does not imply that motor cortical neurons that typically show a broader tuning all belong to the same ‘population’ as in terms of clusters. We now discuss this limitation and suggest next modeling steps to address this shortcoming in the new subsection “Model limitations and future directions” of the Discussion.

4. In a different type of the delayed center-out task (e.g., Kaufman et al., eNeuro, 2016), the variance of reaction time (RT) is due to different durations of the delay period rather than the uncertainty of targets/incomplete information (i.e., whether there's enough time for the preparatory neural states to evolve to an "optimal" initial state). Could the E/I-clustered model explain the RT variance in this case? It would be very valuable to the field if the authors could use their model to examine the potential neural mechanism underlying reaction-time variance in general.

This is an interesting question and we thank the reviewer for highlighting its importance for the field. Kaufmann et al. (2016) show that reaction times increase (their Fig. 3C) in “free-to-forced” trials (where, at the onset of the trial, the monkey has the free choice between two alternative trajectories to reach the goal before one of the routes is blocked) if one of the two route options was blocked at a time point close to the GO signal. Their interpretation is that in trials where the monkey had prepared his movement along the later on blocked trajectory, it needs additional time to re-plan its movement when the block appears. If this block appears around the GO, there is not sufficient time of the delay period left for re-planning and the behavioral response is delayed. This is indeed similar to our task conditions 2 and 3, where the monkey receives full target information only with the GO signal. The authors do not provide a mechanistic explanation at the neuronal level. Indeed, our model of action selection provides a mechanistic explanation for the observed longer reaction times in Kaufmann et al. In cases where the monkey prepared for route A, the attractor for route A would be active. A sudden block of route A requires switching to an attractor that enables route B. Our decision model would require additional time to reach a threshold for action selection and thus should lead to increased reaction times. We now briefly discuss the results by Kaufmann et al. in the context of our model approach in the revised Discussion section.

5. I recognize that it would be tricky to collect new data from a second monkey, but N=2 is the typical practice of monkey neurophysiology studies. The current results would be further strengthened if the authors could repeat them in a second monkey, ideally from existing datasets.

The experiment considered here was carried out in two monkeys (M1 male, M2 female). We have now completed the same experimental data analyses and full model support for the second monkey (M2), specifically in the new Supplementary Fig. S3 and related text. M2 showed a strikingly different behavioral strategy (throughout the training period and during recordings sessions) and produced basically identical reaction times in all three experimental conditions. Our interpretation (see revised Discussion and the new Supplementary Material) is that monkey M2 avoided anticipation of the PS to avoid premature movement initiation (negative reaction times), which would have led to the abortion of the trial. Indeed, anticipatory behavior of M1 led to trial abortions, while this was essentially never the case for M2. This difference was also expressed in the task-dependent dynamics of the FF that, in M2, did not differ across the three different conditions. A further-going interpretation is that M2 did pay less attention to the preparatory stimulus (as indicator of full or partial directional information), which results in a lesser degree of motor cortical activation during the delay period. Such differences in task-solving strategy between individual monkeys are not uncommon. The experimental spike train irregularity (CV2) was again constant in M2 with the same average value (CV ~0.8) as in M1. In our model simulation we could again match the time dependence of FF and CV2 for M2 as well as the similarity of reaction time distributions across experimental conditions. This was achieved with *identical* network parameters, and by adjusting (as a single parameter) the cue-specific stimulation amplitude (lower in M2 during the delay period) of the respective clusters.

In summary, the additional data and model analysis for M2 are in full support of our proposed model that shows sufficient flexibility to accommodate the neural and behavioral differences and commonalities across the two monkeys. This is now outlined and discussed in the new Discussion section.

Minor suggestions:

1. Figure 1: To help readers understand the plots without going back to the main text, I'd suggest adding the full names of PS and RS to figure 1a legend.

Thank you for this helpful comment. In the revised caption of Fig. 1a we provide a brief description of the behavioral task to assist the reader. There we introduce the abbreviations trial start (TS), preparatory signal (PS) and response signal (RS).

2. Figure 1b,c is trying to replicate the decrease of FF after PS as observed in monkey motor cortical neurons. Since the monkey data only involved one type of visual stimulus, could the

authors clarify what the weak and strong stimuli used in the model correspond to in the monkey data?

Fig. 1b,c didactically shows how the metastability is robustly achieved for the E/I network for different stimulus strengths. The brightness of the experimental visual stimuli is fixed across conditions and so is our stimulation strength per cluster across the three different conditions. Of course we cannot know the stimulus amplitude (as input to our model) that corresponds to the effect of the experimental visual stimulus (and any possible effect of attention). Stimulus amplitude is thus the one free parameter that we adapt to reproduce the neuronal and behavioral data measured during the monkey experiment (as in the updated Fig. 5 for monkey M1 and the corresponding additional Supplemental Fig. S3 for monkey M2). Note that for Fig. 5 we determined the stimulus amplitude as 0.1pA (and thus a very weak stimulus in the setting of Fig. 1b,c) to match the experimental firing rates and CV/FF. An even lower stimulus amplitude value were determined for the new Supplemental Fig. S3, monkey M2. Thus, the low range stimulus amplitudes in Fig. 4 are of high relevance and, by interpretation of our results, biologically realistic. See also our response to major point 2 above.

In the revised MS we now indicate the exact stimulus amplitudes in the caption of Fig. 1b,c and in the captions of Fig. 5 and Supplemental Fig. S3. In the main text we now indicate that we are operating our model with low stimulus amplitudes to achieve realistic firing rates and make the relation to our analysis of stimulus-response features for the E and E/I-clustered network topologies in Fig. 4. We were indeed not able to tune the E-clustered model to reproduce any of our experimental results in Fig. 5, Fig. 6 and Supplemental Fig. S3.

3. Line 311: “stimulate” should be “stimulated”.

We thank the reviewer for spotting the typo, which has now been corrected.

4. Figure 6b,c: Could the authors explain why the distributions of reaction time for conditions 2 and 3 are much wider in the model than the monkey data? I'd also suggest adding statistical tests to quantitatively compare the distributions.

We are deliberately using the most simple drift-diffusion decision model where, after RS, the integrated cluster firing rate represents the decision variable that has to cross a fixed threshold. The only two parameters here are the integration time constant and the threshold. The width of the reaction time distributions is thus mainly determined by the across-trial variability of the integrated population spike count. The important aspect in Fig. 6b and c is certainly that in our model ambiguous information leads to parallel and alternating representations of the possible final targets and that this alone suffices to explain the (significantly) longer reaction times in experimental conditions 2 and 3. The same underlying mechanism can explain longer reaction times in Kaufmann et. al. (2016), see major comment 4 above.

5. Does the model capture other behavioral variability in addition to the reaction-time variability, such as the end-point variance of reaches? It is also not clear to me if the replication of the behavioral variability is due to adding the inhibitory clusters or is intrinsic to any network model

(i.e., both E-clustered and E/I-clustered models)? Could the authors compare the behavioral predictions of both models?

In our experiment the monkeys performed free arm reaching movements. The behavioral event data was monitored (i.e. the time point when the monkey left the central starting point and the time point when he touched the target) but not the hand trajectory. From this data we can compute reaction times and movement duration.

We were not able to tune the E-clustered network to reproduce average firing rates, FF and CV2. Matching the rather low average motor cortical firing rates and at the same time observe a reduction in the FF is not possible because, as explained above in response to major comment 2, low stimulus amplitudes are required to match the observed average firing rates in the motor cortex (Fig. 5), which result in an increase of the average FF (Fig. 1). Conversely, if we use stronger stimuli, the average firing rates can only assume too large values that do not match the experimental firing rates (Fig. 4). We did not further attempt to match reaction times in the E-clustered model with non-matching neuronal statistics. In Mazzucato et al., 2019 the authors were able to match reaction times in data recorded from gustatory cortex of rats using an E-clustered model, but did not regard neuronal statistics.

6. Related to my last comment, the reaction time prediction doesn't seem to be unique to the E/I-clustered network: my understanding is that for any network with competing ramping activity towards multiple decision boundaries, there would be an uncertainty/evidence-dependent variance in reaction time. Could the authors rule out this possibility? Or alternatively, could the authors show the unique strengths of their model in terms of predicting behavior? For instance, as I mentioned in previous comments, maybe the authors could test whether their model is capable of predicting behavioral parameters in addition to reaction-time variance and outperforms other E/I-balanced networks.

We cannot rule out the possibility that other models with competition among different targets in the uncertain experimental condition might be able to generate similar reaction times (see above). However, the fact that in the E/I-clustered model we observe robust switching dynamics over a large range of network parameters ensures robust competition and does not require fine-tuning of network parameters as in the E-clustered model.

Reviewer #2 (Remarks to the Author):

The standard balanced network model produces an irregular spike train observed in vivo. To further explain slow spike rate fluctuations and trial-to-trial variability in the cortex, balanced network models with clustered excitatory connections have been proposed. Yet, these models require fine-tuning. In addition, the spike rate becomes extremely high when trial-to-trial variability is matched to experimental data. Rostami et al implemented and analyzed a balanced network model with excitatory and inhibitory clustering. They have shown the model explains task-related dynamics in the primate motor cortex well, including a decline in trial-to-trial

variability across task conditions. These findings are interesting and important, yet multiple issues need to be addressed, as listed below

We thank the reviewer for the positive account of our manuscript and the constructive and very helpful comments! We apologize for the severely delayed revision due to personal reasons.

Below we respond in detail to all major and minor comments that we have all addressed in a thorough revision of our manuscript. In summary, we provide an additional analysis of the E+I type network topology as proposed in Litwin-Kumar et al. in Supplemental Figure S1 that showed the same behavior and requires the same fine-tuning as the E-clustered network. We added the mean-matched Fano factor analysis (Supplementary Figure S2) that shows the same result as our combined analysis of FF and CV2. Moreover, we repeated our analyses and simulations for a second monkey that performed the identical task (Supplementary Figure S3); our analysis now also includes the task-related firing rates (revised Fig. 5 and Supplementary Figure S3). Furthermore, we have worked extensively on including our model as a part of Nest Simulator library to make our model available to the community.

All text changes in the Abstract, Introduction, Results and Methods have been marked in blue color with the exception of minor corrections (such as typos). We strongly revised the Discussion accommodating the suggestions of both reviewers. We therefore marked the complete Discussion in blue text color without indicating deleted text (of the original version) to ease reading.

Major

1) *The authors claim that the EI-clustered network is a novel architecture (P.4 L.69). Yet, the EI-clustered network has been reported in Litwin-Kumar et al. (Fig8) to explain trial-to-trial variability in inhibitory neurons. I agree that analysis in Litwin-Kumar is limited, and authors have reported new features of the network. Yet, this should be clarified in the manuscript.*

We are aware of the analysis in Fig. 8 of Litwin-Kumar et al. The clustered network analyzed there (termed “E+I”) was derived from the E-clustered network by introducing *unidirectional* excitatory E→I connections from each of the previously defined excitatory clusters to one new cluster of inhibitory neurons. By this, the authors could demonstrate a temporary reduction of the Fano factor not only in the excitatory but also in the inhibitory population upon stimulation of the excitatory cluster (Fig. 8C). Note, however, that an inhibitory cluster is defined solely by its directed input from an excitatory cluster. There is neither reciprocal I→E connectivity between these clusters (rather each excitatory cluster received input from a random selection from all inhibitory neurons and thus no local balanced input to neurons in the excitatory cluster) nor increased reciprocal I↔I connectivity (and thus no balanced input to neurons in the inhibitory cluster).

Our definition of the E/I-clustered network is thus different. With increasing a single parameter J_{E+} we can “slide” the network from the standard balanced network ($J_{E+} = 1$) to an

increasingly stronger E/I-clustered network, where J_{E+} is coupled to the inhibitory cluster strength J_{I+} through equation 13.

For the revised MS we have now implemented the E+I clustered network of Litwin-Kumar et al. and performed the same calibration analysis as in Fig 2b/d. As expected, the result is essentially the same as for the E-clustered network where the FF, as a proxy to metastability, shows higher values only for the same small parameter range. This result is now included in the supplemental material (Supplementary Fig. S1) together with a description of the E+I network architecture. We refer to this additional analysis in the corresponding Results section. We now briefly discuss the E+I network as outlined in Litwin-Kumar et al. in the Discussion section.

2) Metastability is not explained well, despite its importance in this paper. The introduction should be expanded for non-experts.

We thank the reviewer for this important comment. We now provide an introductory description and definition of metastability in the Introduction that should support the reader in following our line of arguments and interpreting our results.

3) Spike rates (SR) need to be analyzed in more detail. It is important to show SR is matched besides FF and CV2 to claim the model captures task-related dynamics. Most figures and analyses do not show SR and PSTH (e.g., Fig1 and 5).

Fig 4 shows that spike rates (SR) need to be extremely high in the E-clustered network to reduce FF, while it is not the case for the E-I network. Do both FF and SR in the E-I network match the data (text only mentions that FF matches the data)?

We had adjusted the stimulus amplitudes to match the average firing rates but had not shown them in the original version of our MS. In the revised version we have now added the firing rate estimates in the monkey recordings and model simulations to Fig. 5 (and Supplementary Fig. S3). The average firing rates of the E/I model simulations generally match those in the monkey. They are in a range of approx. 10 - 25 Hz. Correspondingly, the stimulus amplitudes used in our model are in the very low range of the tested amplitudes in Fig. 4. The E-clustered network fails to reproduce these firing rates in a stable manner and cannot account for a reduction of the FF in this stimulus/firing rate regime, as expected from Fig. 4. We therefore failed to find any parameter set for the E-clustered network that could reproduce the experimental physiology and behavior even remotely.

4) P.9 L.185 “Local balance of excitation and inhibition facilitates attractor dynamics and maintain spiking irregularity”. This part of the paper beautifully explains how the E-clustered network results in mean driven regime to produce regular spiking. In contrast, the E-I clustered network maintains under a fluctuation-driven regime to keep the irregular spiking. But as far as I understand, it says nothing about “facilitates attractor dynamics”. Please remove it or explain what it means.

We agree with the reviewer that in the original MS this was not explained explicitly. We have now added a summarizing paragraph at the end of the respective section to better explain the effect of presence or absence of balancing inhibition on the (spontaneous) switching dynamics of individual clusters between activated and inactivated state and hence on the desired attractor dynamics where the network is expected to cycle through different metastable states that are represented by different (combinations of) activated clusters, and how on the contrary the mean-driven input regime in the E-clustered network lead to longer (and potentially very long) dwell times of an individual self-exciting cluster with high firing rates. In the revised Discussion (first subsection) we briefly discuss how in the E-clustered network robust metastability is hampered and how the E/I topology achieves metastability robustly.

5) It is critical to rule out whether declining FF is due to rising SR. Show SR and mean matched FF (Figs 1 and 5).

We have now conducted the additional analysis of the mean matched FF for Fig. 5 following the method proposed in Churchland's work. The results are shown in Supplementary Fig. S2. We find that the time-resolved FF follows our previous analysis with unrestricted count distribution; if at all, the FF reduction after PS is even stronger in the mean-matched estimate. This additional control is reassuring, indicating that the reduction in FF is not caused by an increasing firing rate.

Our rationale for not estimating the mean-matched FF in the original MS version is that we co-analyze FF and CV2. A rising firing rate can have a regularizing effect on the neuron's spiking (due to absolute and relative refractoriness) and thus a decreasing effect on the irregularity measure (CV and CV2 respectively), which in turn can reduce the FF. The fact that the (average) CV2 stays constant across trial time indicates that the reduction of the excess Fano factor (excess in the sense that the FF is considerably larger than the squared CV2) in trial time is not an effect of a change in the neurons' firing rate.

We now refer to the mean-matched FF analysis as an additional control for (high) firing rate effects on the Fano factor in the main text and provide method details in the Supplemental material along with the Suppl. Fig. S2. We also briefly reflect on the drawbacks of the mean-matched FF method with respect to a change in neuron composition and possible emphasis of the estimation bias.

Note, that in the motor cortex average and peak firing rates are typically considerably smaller than in sensory cortices. If we, for example, compare our data to the data in Fig. 4 of Churchland et al. (2010), then we find counts of up to 10 within a 50ms window, which corresponds to rates of up to 200Hz. In our data we hardly ever find firing rates that exceed approx. 60-70Hz in dynamic estimates of the rates. Thus, firing rate peaks in our data correspond to the upper bound of 5 (corresponding to 100Hz) for the count distributions used for the mean-matched FF analysis in Fig. 4 of Churchland et al. (2010). If we assume that the biophysical properties of neurons in both cortices are similar, then the influence of firing rate on the FF is likely much less effective in the analysis of motor cortical data.

Minor

1) *Figure 1. It looks odd that FF declines before the task onset. Use a causal sliding window.*

We decided to use a centered observation window rather than a causal window so that the center matches the time scale of the experiment and this is consistent with all our analyses we performed in previous publications. Thus we would like to stick to the chosen centering. If we used a causal kernel, then we would have firing rate and FF delayed with respect to the time-line of the task. We now explicitly pointed out that we are using a centered window and therefore, the FF curve can start to decrease before the PS trigger event (max. 200ms).

2) *In Figure 5f, there is no FF decline at the task onset, unlike in the data (5c). Please comment on this.*

Yes, the experimental data in Fig 5f (monkey M1) shows an initial short-lived decline in FF in response to the onset of the preparatory stimulus (PS) after which it “recovers” to the more steady state in all three conditions.

We do have a fairly good hypothesis about the mechanistic cause for this initial reduction following the onset of the PS. In a previous study we proposed spike frequency adaptation (SFA) to account for a transient reduction in FF (Farkhooi et al., 2013) after the onset of a stimulus. We had established this in simulation and in analytic treatment. However, this effect is only transitory, recovering with the time scale of the SFA conductance, and cannot explain the maintained reduction of the FF that, as outlined in the present MS, is most likely an effect of the attractor dynamics. Our (very) preliminary results indicate that, indeed, the combination of both mechanisms can reproduce the experimentally observed variability dynamics including the initial transient reduction after onset of the target stimulus (due to SFA) and a maintained reduction at a plateau level throughout the preparatory phase (due to the attractor dynamics).

Furthermore, monkey M2 displays a decisively different behavioral strategy (by largely ignoring the PS and reacting to the RS) resulting in almost equal (long) reaction times in all three experimental conditions while the FF reduction does not reflect the three conditions. The time course of FF in monkey M2 does not clearly show the initial transient FF reduction. We captured the behavior of the FF in monkey M2 in our E/I-clustered network model by a reduced input stimulation (stimulus amplitude is reduced by half compared to the model stimulation for M1) of clusters with the target stimulus (PS). This again fits the SFA hypothesis where a reduced input drive of neurons will result in weak responses and a correspondingly weak SFA conductance, which would result in only a very weak stimulus onset effect on FF reduction.

While the analysis of the concerted effect of clustering and SFA on attractor dynamics and FF dynamics is beyond the scope of the present MS, we now put forward our hypothesis in the Discussion and will follow this line of research in the near future.

3) *The model in Figs 5D-F & 6 predicts competition between three clusters tuned for different targets in the triple target condition. If you happen to have simultaneously recorded data, do you see evidence of “competition” among neurons with different tuning (i.e., different clusters)?*

Indeed, it would be very interesting to analyze such competition of neurons of different tuning as a means of identifying possible clusters. Ideal for this analysis would be a data set of many simultaneously recorded neurons. Unfortunately, in the present data set, we deal with acute recordings of very few (1-3) simultaneously recorded neurons. We'd therefore like to address the identification of neuron clusters in future data-analytic work with collaborative partners.

4) Daie et al., NN, 2021 uncovered a cluster-like structure in the frontal cortex of mice performing a delayed response task. It may be worth citing as it supports the clustered network in a similar task.

We thank the reviewer for this citation that had escaped our attention. We have now cited this work in our revised Introduction.

REVIEWERS' COMMENTS

Reviewer #1 (Remarks to the Author):

Overall the authors have sufficiently addressed most of the comments in the previous review, and the clarity of their rationale and methods has largely improved. The new results from a second monkey demonstrated in Fig. S3 helped strengthen the replicability of the findings despite monkey 2 using a different behavioral strategy in movement planning. Although the neural patterns and “behavioral” output of the E/I-clustered neural network did not resolve certain discrepancies with the monkey data, they provided insights into how E/I clustering could enhance the robustness of a network to maintain winnerless attractor dynamics and its relationship with reaction time variability. The detailed comparison to other E/I-balanced models helped with a better understanding of the relevance and limitations of the current model as well as directions for future models. I have a few remaining suggestions for the revised manuscript:

1) The authors emphasized the biological realism of their neural network compared to others. As the cluster strength was one key parameter that was varied for characterizing the E-clustered and E/I-clustered models, I was wondering about its biological relevance: was it an arbitrary parameter in the model to show the possibility for the network to display different types of dynamics (no attractor vs. winnerless competition vs. winner takes all), or does it actually match biological networks that show different types of dynamics? Could biological networks achieve metastability without the need for a wide range of cluster strengths?

2) Mismatch between neurophysiological FF dynamics and the model output in Figure 5c,f: in the monkey data, delta FF dramatically decreased for both conditions 2 and 3 around RS, but there's no obvious decrease of the FF for condition 2 in the model. Moreover, while FF for all three conditions reduced to around the same level after RS in the monkey data, it wasn't the case for the model. Because matching FF in the model to neurophysiological data is a critical claim made in the manuscript, could the authors either explain this discrepancy in the main text, or further improve their model to match the neurophysiological FF dynamics over the course of action selection?

3) The complementary nature of this work and Hennequin et al 2014 could be an important point to connect to past work and also inform future studies. Thus, I would recommend the authors converting their nicely written responses to my major point 1 into a formal section in the supplementary material.

4) Minor points:

Figure1a: maybe also provide delta FF to directly echo results in figure 4.

Line 126: “4.000” and “1.000” should be “4,000” and “1,000”. Same for line 165 and a few other places.

Line 218: “until at about $t = 350\text{ms}$ *when* one cluster...”

Line 592: “unrestricted movements, *and* their hand trajectories were not recorded”

Reviewer #1 (Remarks on code availability):

The README file is clearly written for setting up the code and accessing the uploaded data to reproduce figures as presented in the manuscript. Its applicability to other datasets remains to be determined.

Reviewer #2 (Remarks to the Author):

All my concerns/comments have been addressed. I think it is ready for publication.

Reviewer #1 (Remarks to the Author):

Overall the authors have sufficiently addressed most of the comments in the previous review, and the clarity of their rationale and methods has largely improved. The new results from a second monkey demonstrated in Fig. S3 helped strengthen the replicability of the findings despite monkey 2 using a different behavioral strategy in movement planning. Although the neural patterns and “behavioral” output of the E/I-clustered neural network did not resolve certain discrepancies with the monkey data, they provided insights into how E/I clustering could enhance the robustness of a network to maintain winnerless attractor dynamics and its relationship with reaction time variability. The detailed comparison to other E/I-balanced models helped with a better understanding of the relevance and limitations of the current model as well as directions for future models. I have a few remaining suggestions for the revised manuscript:

1) The authors emphasized the biological realism of their neural network compared to others. As the cluster strength was one key parameter that was varied for characterizing the E-clustered and E/I-clustered models, I was wondering about its biological relevance: was it an arbitrary parameter in the model to show the possibility for the network to display different types of dynamics (no attractor vs. winnerless competition vs. winner takes all), or does it actually match biological networks that show different types of dynamics? Could biological networks achieve metastability without the need for a wide range of cluster strengths?

According to our simulation results a cortical E/I network actually does not require a wide range of cluster strengths to exhibit metastability; rather, metastability is achieved over a large range of this parameter, meaning that a network with a fixed parameter within this range will experience winnerless competition. This also means that metastability in this network architecture is robust against variation in (relative) cluster strength due to homeostatic regulation or other types of modulations that affect synaptic strength, background input, excitability or connectivity. Moreover, in our case for both monkeys we could adapt the E/I network with the very same single-valued cluster strength parameter.

Unfortunately, to our knowledge there are no quantitative estimates of in vivo cluster strength in cortical networks and, thus, we parameterized our network such that it reproduces average in vivo firing rates and second order spiking statistics. In the present study we have chosen a fixed connectivity and varied synaptic strength to define cluster strength, and we fixed the relative strengths of excitatory and inhibitory clustering by R_J (eqn. 14). Litwin-Kumar and Doiron (2012) had introduced the E-clustered network by varying the connection probabilities between and within clusters. Both approaches maintain the same average input current to single neurons, but they differ in the variance due to distinct correlation structures resulting from changes in the number of participating pre-synaptic neurons. In a biological network a combination of both - local vs. global connectivity and synaptic weights - may constitute a cluster.

In the living brain, we may further expect that clusters are formed by plasticity and through learning resulting in a certain range of individual cluster strengths (each effectively representing attractor depth). This would allow e.g. to represent probabilities or priors. Local balance and network stability may be maintained by homeostatic means. We may further hypothesize that neuromodulation by e.g. biogenic amines or endocrine factors can change synaptic strengths and plays a role in controlling different types of dynamics on short or long time scales. Thus, we hypothesize that the living brain explores at least a moderate range of cluster strength.

Learning or modulation of cluster strength by either varying e.g. synaptic strength or connectivity is beyond the scope of the present manuscript; we therefore decided to not further increase the length of our Discussion section but rather leave this to future studies.

2) Mismatch between neurophysiological FF dynamics and the model output in Figure 5c,f: in the monkey data, delta FF dramatically decreased for both conditions 2 and 3 around RS, but there's no obvious decrease of the FF for condition 2 in the model. Moreover, while FF for all three conditions reduced to around the same level after RS in the monkey data, it wasn't the case for the model. Because matching FF in the model to neurophysiological data is a critical claim made in the manuscript, could the authors either explain this discrepancy in the main text, or further improve their model to match the neurophysiological FF dynamics over the course of action selection?

In the present study we focused on the preparatory phase and the modulation of the FF as a result of the preparatory stimulus that determines the amount of cued target information. The RS presents a second behavioral relevant stimulus and the neuronal dynamics that follows falls into the actual movement phase. In our network simulation for monkey M2 where we step up the small stimulus amplitude from PS to RS, we achieve a qualitatively and quantitatively comparable reduction of the FF, both in simulation and in the monkey data. Our simulation for M1, however, does not match the empirical rapid FF reduction in the neuronal data in the case where only RS provides complete data (Conditions 2 and 3). Improving the model fit for M1, however, would from our point of view require the model extension towards a model that encodes movement during execution as mentioned below in point 3 and now outlined in the Supplementary information and the assumption that stimulus quality changes with RS (i.e. not only a release from input) such that a novel input will lead to a reduction in FF similar to M2. Additionally we may again hypothesize that cellular adaptation plays a role as indicated by the again rather rapid reduction. We plan to address the task of matching FF during actual behavior in a future model study, possibly including additional experimental data sets. At this stage we would like to refrain from discussing possible multifaceted reasons for the mismatch as, at this point, this would be purely speculative.

3) The complementary nature of this work and Hennequin et al 2014 could be an important point to connect to past work and also inform future studies. Thus, I would recommend the authors converting their nicely written responses to my major point 1 into a formal section in the supplementary material.

We followed the reviewer's suggestion and, based on our replies to major point 1, compiled the new formal section "Possible model extension to accommodate action selection and explicit motor control signals." in the Supplementary Information.

4) Minor points:

Figure 1a: maybe also provide delta FF to directly echo results in figure 4.

We decided to keep Figure 1 as is as we think it is also relevant to show the absolute quantitative measure of FF for the experimental data (a) and the two model variants (b,c). The change in FF in Figure 4 is echoed in Fig. 5 (and supplemental Fig. 3) for the experimental data.

Line 126: "4.000" and "1.000" should be "4,000" and "1,000". Same for line 165 and a few other places.

Done

Line 218: "until at about t = 350ms *when* one cluster..."

Done

Line 592: “unrestricted movements, *and* their hand trajectories were not recorded”

Done

Reviewer #1 (Remarks on code availability):

The README file is clearly written for setting up the code and accessing the uploaded data to reproduce figures as presented in the manuscript. Its applicability to other datasets remains to be determined.

We have further improved the README and code documentation. Additionally, the ‘EI clustered circuit model’ has now also been submitted, reviewed and added to the NEST simulator as reviewed example code with documentation. This may further support testing the applicability to other datasets. The link to the NEST implementation and documentation

https://nest-simulator.readthedocs.io/en/latest/auto_examples/EI_clustered_network/index.html

is provided in our repository. As suggested by the journal we now provide the code repository with a DOI as outlined in the Code Availability statement.